# Direct cell–cell contact between mature osteoblasts and osteoclasts dynamically controls their functions in vivo

Masayuki Furuya[1,2,3], Junichi Kikuta [1,2], Sayumi Fujimori[2], Shigeto Seno [4], Hiroki Maeda[5], Mai Shirazaki[1,2], Maki Uenaka[1,2], Hiroki Mizuno[1,2], Yoriko Iwamoto[1,2], Akito Morimoto[1,2], Kunihiko Hashimoto[1,2,3], Takeshi Ito[6], Yukihiro Isogai[7], Masafumi Kashii[3], Takashi Kaito [3], Shinsuke Ohba[8], Ung-il Chung[8], Alexander C. Lichtler[9], Kazuya Kikuchi[5], Hideo Matsuda [4], Hideki Yoshikawa[3] & Masaru Ishii[1,2]

Bone homeostasis is regulated by communication between bone-forming mature osteoblasts (mOBs) and bone-resorptive mature osteoclasts (mOCs). However, the spatial–temporal relationship and mode of interaction in vivo remain elusive. Here we show, by using an intravital imaging technique, that mOB and mOC functions are regulated via direct cell–cell contact between these cell types. The mOBs and mOCs mainly occupy discrete territories in the steady state, although direct cell–cell contact is detected in spatiotemporally limited areas. In addition, a pH-sensing fluorescence probe reveals that mOCs secrete protons for bone resorption when they are not in contact with mOBs, whereas mOCs contacting mOBs are non-resorptive, suggesting that mOBs can inhibit bone resorption by direct contact. Intermittent administration of parathyroid hormone causes bone anabolic effects, which lead to a mixed distribution of mOBs and mOCs, and increase cell–cell contact. This study reveals spatiotemporal intercellular interactions between mOBs and mOCs affecting bone homeostasis in vivo.

[1] Department of Immunology and Cell Biology, Graduate School of Medicine & Frontier Biosciences, Osaka University, Osaka 565-0871, Japan. [2] Department of Cellular Dynamics, WPI-Immunology Frontier Research Center, Osaka University, Osaka 565-0871, Japan. [3] Department of Orthopaedics, Graduate School of Medicine, Osaka University, Osaka 565-0871, Japan. [4] Department of Bioinformatic Engineering, Graduate School of Information Science and Technology, Osaka University, Osaka 565-0871, Japan. [5] Department of Material and Life Sciences, Graduate School of Engineering, Osaka University, Osaka 565-0871, Japan. [6] Laboratory for Pharmacology, Pharmaceutical Research Center, Asahi Kasei Pharma Corporation, Tokyo 101-8101, Japan. [7] Medical Affairs Department, Pharmaceutical Business Administration Division, Asahi Kasei Pharma Corporation, Tokyo 101-8101, Japan. [8] Division of Clinical Biotechnology, Center for Disease Biology and Integrative Medicine, The University of Tokyo, Tokyo 113-0033, Japan. [9] Department of Reconstructive Sciences, School of Dental Medicine, University of Connecticut Health Center, Farmington, CT 06030, USA. Correspondence and requests for materials should be addressed to M.I. (email: mishii@icb.med.osaka-u.ac.jp)

Bone undergoes continuous remodeling throughout life. The bone remodeling process, beginning with bone resorption by osteoclasts followed by bone formation by osteoblasts, takes place asynchronously throughout the skeleton at anatomically distinct sites known as basic multicellular units (BMUs)[1,2]. Tight control of bone remodeling at the BMU level is critical for maintaining bone homeostasis in response to structural and metabolic demands. Bone remodeling is strictly controlled through a complex cell communication network with signals between osteoblast and osteoclast lineage cells at each BMU[3,4]. Therefore, it is essential to understand the spatial-temporal relationship and interaction between osteoblasts (including their mesenchymal pre-osteoblastic precursors) and terminally differentiated osteocytes and osteoclasts (including their monocytic precursors) in vivo. In particular, it remains controversial whether these cell types physically interact with each other, as bone resorption and formation occur in physically and temporally discrete units of cellular activity[1,2].

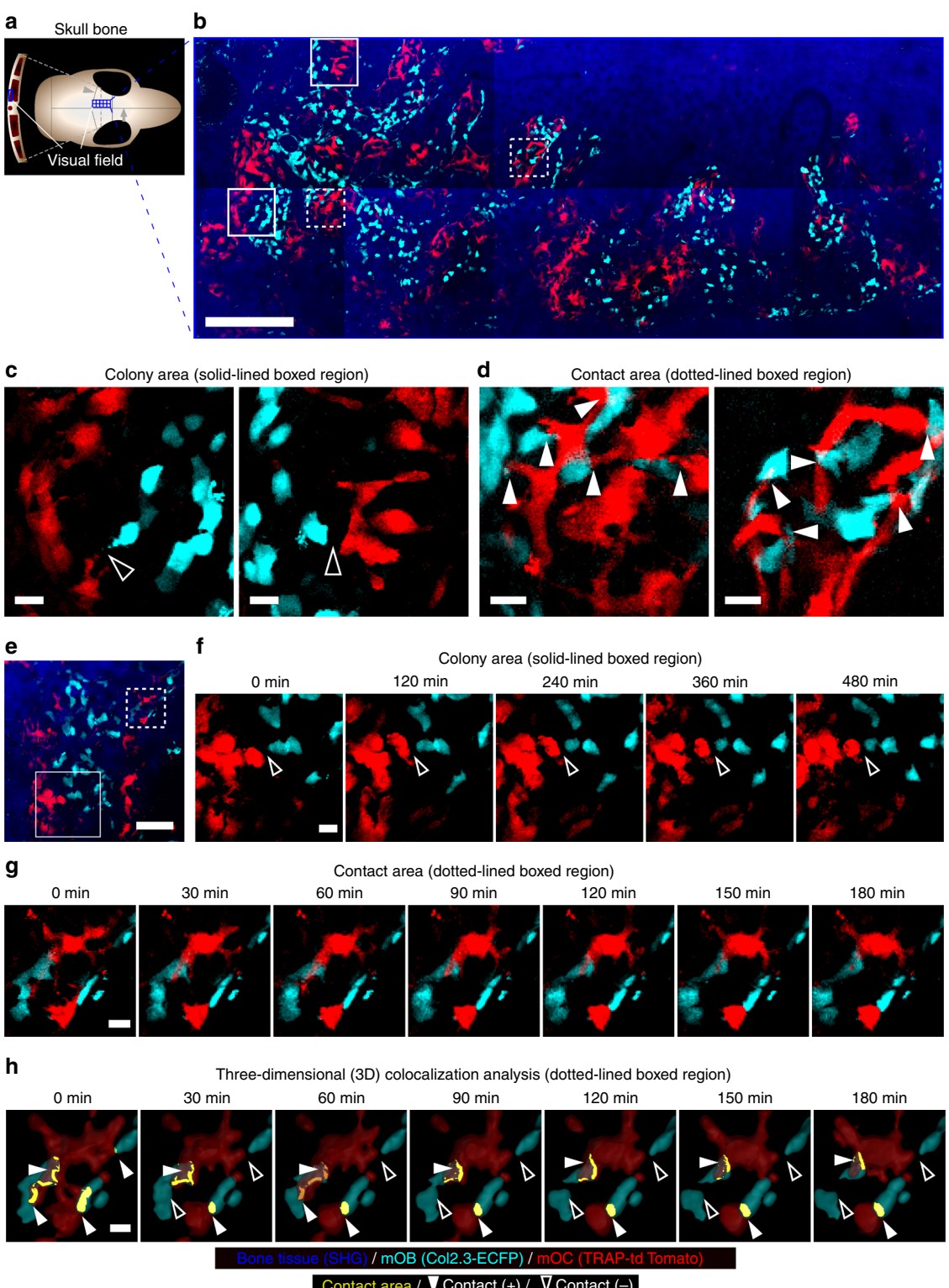

Over the past two decades, intravital two-photon microscopy has launched a new era in the field of biological imaging[5,6]. The near-infrared excitation laser for two-photon microscopy can penetrate thicker specimens, making it possible to acquire spatial-temporal information of living cells and visualize the behavior and interaction of living cells within tissues and organs. Indeed, intravital two-photon microscopy enables observation of living cells within bone tissues in vivo[7–10].

In this study, we investigate the communication between mature osteoblasts (mOBs) and mature osteoclasts (mOCs) in vivo. Using two-photon microscopy, mOBs and mOCs are visualized at the same time in living skull bone tissues from transgenic mice that express enhanced cyan fluorescent protein (ECFP) driven by the type I collagen promoter in mOBs and tdTomato (a red fluorescing protein), under the control of the tartrate-resistant acid phosphatase (TRAP) promoter in mOCs. This simultaneous visualization reveals that mOBs and mOCs mainly occupy discrete territories in the bone marrow in the steady state, although direct cell-to-cell contact exist in a spatiotemporally limited manner. A novel fluorescent probe developed to detect bone-resorptive proton secretion demonstrates that direct contact with mOBs inhibit bone resorption by mOCs. In addition, we show that these modes of interaction are dynamically altered according to bone homeostatic conditions; intermittent administration of parathyroid hormone (PTH), which leads to bone formation, increases the frequency of the direct physical interaction between these two cell types.

## Results

**Generation of reporter mice expressing ECFP in mOBs**. To simultaneously visualize mOBs and mOCs in vivo, we generated transgenic reporter mice that expressed differing fluorescent proteins in the cytosol of mOBs and mOCs. Previously, we generated reporter mice expressing tdTomato, a red fluorescent protein, in the cytosol of mOCs[9]. Here we generated fluorescent reporter mice expressing ECFP in mOB cytosols. We used a transgene-expressed ECFP driven by the 2.3 kb fragment of rat type I collagen α (1) promoter (Col1a1*[2.3]) for specifically labeling mOBs, which we call Col2.3-ECFP hereafter (Supplementary Fig. 1a)[11,12]. Using bone tissue sections from these mice, immunohistochemistry analysis provided confirmation that ECFP fluorescence was expressed in the endosteal and trabecular osteoblasts, and ECFP-positive cells expressed alkaline phosphatase (ALP) (Supplementary Figs. 1b, c). The time-dependent changes of ECFP fluorescence in bone marrow stromal cell (BMSC) cultures derived from Col2.3-ECFP mice were evaluated. ECFP fluorescence was localized in mineralized nodules, which facilitated detection (Supplementary Figs. 1d, e). In addition, quantitative reverse-transcription PCR analysis of BMSC cultures of Col2.3-ECFP mice revealed that ECFP expression coincided

with those of osteocalcin but not Col1 or ALP (Supplementary Fig. 1f), confirming the specific expression of ECFP in fully differentiated osteoblasts. Using a modified intravital two-photon bone imaging technique[7–10], we visualized ECFP-positive mOBs (Supplementary Fig. 1g), which have been shown to move slowly.

**Simultaneous visualization of mOBs and mOCs in living bones**. We generated double fluorescent reporter mice expressing tdTomato in mOCs and ECFP in mOBs by crossing TRAP-tdTomato with Col2.3-ECFP mice, forming Col2.3-ECFP/TRAP-tdTomato mice. Using bone tissue sections from these mice, ECFP-positive mOBs and tdTomato-positive mOCs were observed along the bone surface (Supplementary Figs. 1h–j). Intravital bone imaging of skull bone tissues in Col2.3-ECFP/TRAP tdTomato mice provided simultaneous visualization of mOBs and mOCs; the imaging results suggested direct mOB–mOC contact in vivo (Fig. 1a, b).

In wide views of the endosteal surface of skull bones, mOBs and mOCs appeared to be distributed separately (Fig. 1c, open arrowheads), although some direct mOB–mOC contacts could be seen only in a spatiotemporally limited manner (Fig. 1d, filled arrowheads). Time-lapse image analyses showed that mOBs and mOCs appeared separated at certain distances (~ 10 μm) between these cell types (Fig. 1e, f and Supplementary Movie 1). However, several mOCs in contact with mOBs displayed dendritic shapes, with synapse-like projections toward mOBs (Fig. 1e, g and Supplementary Movie 1).

We further investigated the extent of direct cell–cell contact between mOBs and mOCs, via quantitative three-dimensional (3D) colocalization analysis using a modified version of a previous method[13]. Briefly, the original images were processed with the aid of a Sobel filter to enhance the cell edges (Supplementary Figs. 2a, b). Then, the cell surfaces of all cyan-positive and red-positive cells were segmented with the aid of Imaris software (Supplementary Fig. 2c). Finally, areas of colocalization of the cyan and red color voxels were automatically detected (Supplementary Fig. 2d and Fig. 1h). This allowed us to estimate the extent of direct cell–cell contact between mOBs and mOCs, with the limitation that some contacts may have exited the visual field during intravital time-lapse imaging.

These imaging results revealed that mOBs and mOCs mainly occupied discrete territories and some direct cell-to-cell contact was detected in spatiotemporally limited areas.

**Contact with mOBs inhibits bone-resorbing activity of mOCs**. To examine the functional significance of direct cell–cell contact between these two cell types, we further analyzed the bone-resorptive capacity of mOCs that contacted mOBs using a pH-sensing chemical probe, pHocas-3, which we developed recently[14] (Fig. 2a). This probe, which emits green fluorescence in acidic

**Fig. 1** Simultaneous visualization of intact mOBs and mOCs in living bones. **a** Schematic representation of mouse skull bone. Each small boxed region represents a single visual field and each tiling image contains 10 contiguous boxed regions. **b** A representative, intravital, two-photon, microscopic tiling maximum-intensity projection (MIP) image of skull bone tissues from Col2.3-ECFP/TRAP-tdTomato mice held under control conditions. Cyan, mOBs expressing Col2.3-ECFP; red, mOCs expressing TRAP-tdTomato; blue, bone tissues (second harmonic generation, SHG). Scale bar, 300 μm. **c**, **d** Magnified images from **b** reveal two representative modes of communication between mOBs and mOCs. Scale bar, 20 μm. **c** Representative images of a colony region (the region outlined in **b**). Open arrowheads represent mOBs and mOCs that are separated. **d** Representative images of a contact area (the region delineated by dotted lines in **b**). The filled arrowheads indicate areas of direct mOB–mOC contact. **e**–**g** Intravital two-photon microscopy time-lapse MIP images of skull bone tissues from Col2.3-ECFP/TRAP-tdTomato mice held under control conditions. **e** A representative MIP image captured at 0 min. Scale bar, 100 μm. **f** Magnified images of a colony area (the region outlined in **e**) captured at 0, 120, 240, 360, and 480 min. Scale bar, 20 μm. **g** Magnified images of the contact area captured at 0, 30, 60, 90, 120, 150, and 180 min (the region delineated by dotted lines in **e**). Scale bar, 15 μm. **h** 3D colocalization analysis of the images shown in **g**. The contact area is the region of colocalization of mOBs and mOCs and is shown in yellow. Scale bar, 15 μm

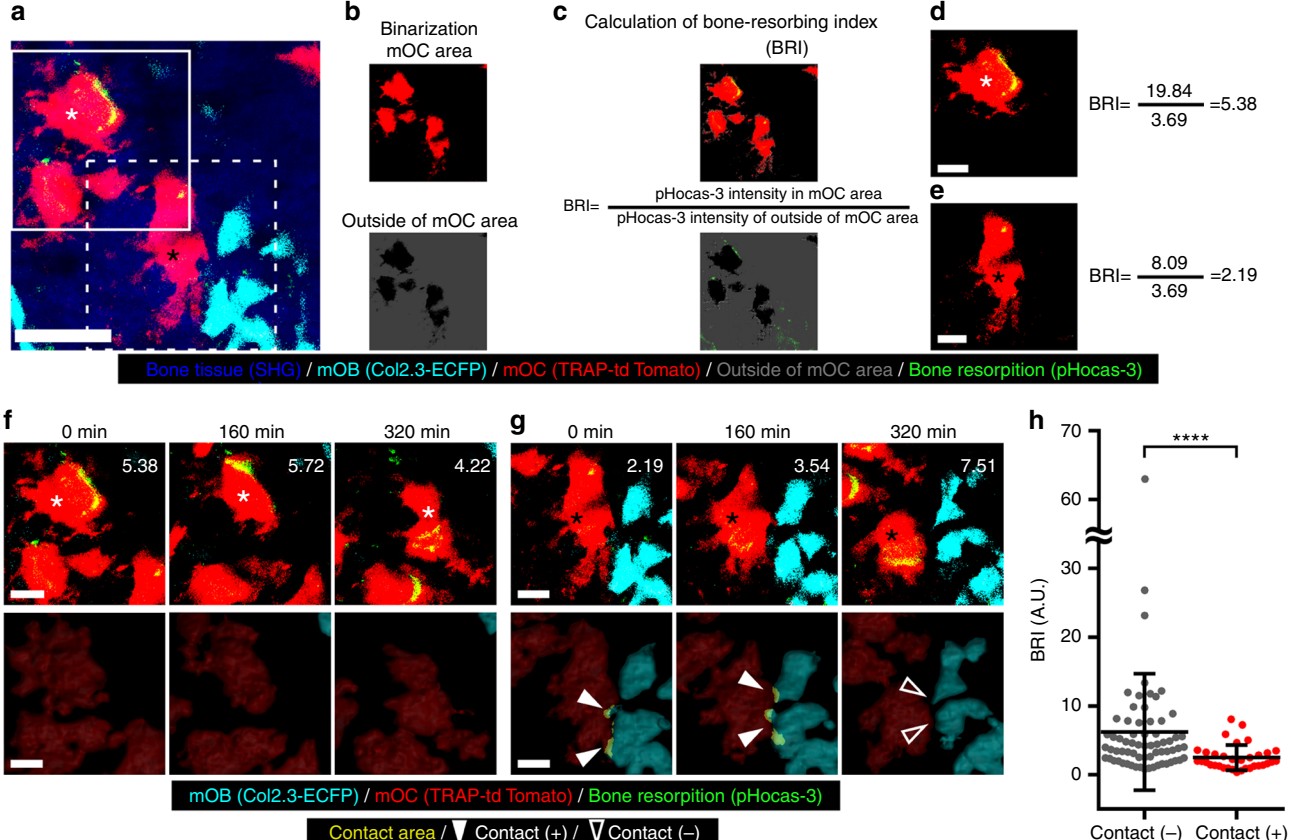

**Fig. 2** Direct contact with mOBs inhibits the bone-resorbing activity of mOCs. **a** A representative MIP image of bone-resorptive activity in skull bone tissue of a Col2.3-ECFP/TRAP-tdTomato mouse injected with a pH-sensing chemical probe, pHocas-3. Green, pHocas-3; cyan, mOBs expressing Col2.3-ECFP; red, mOCs expressing TRAP-tdTomato; blue, bone tissues (SHG). Scale bar, 50 μm. **b**, **c** Assessment of mOC bone-resorbing activity. **b** Areas containing mOCs were automatically binarized from the original images. Red, mOC areas; gray, regions outside the mOC areas. **c** Mean pHocas-3 fluorescence intensities were measured inside (pHocas-3 signal) and outside the mOC areas (pHocas-3 noise). The bone resorbing index (BRI) was the ratio of the pHocas-3 signal to the pHocas-3 noise. **d**, **e** Images processed for BRI calculations (**d** for the mOC indicated with white asterisk in the outlined region of **a**, and **e** for the mOC indicated with black asterisk in the region delineated with a dotted line in **a**). Scale bar, 20 μm. **f**, **g** Magnified MIP images from the region outlined in **a** and **f**, and the region delineated by the dotted line in **a** and **g**, captured at 0, 160, and 320 min (upper panels). The 3D images yielded by colocalization analysis (bottom panels). Scale bar, 20 μm. The contact areas were those where mOBs and mOCs colocalized and are shown in yellow. The filled arrowheads show areas of mOB–mOC contact. The open arrowheads indicate separated mOBs and mOCs. The actual BRI values are shown to the right of the images. **h** BRI of mOCs in contact, or not, with mOBs. Snapshot MIP images were collected from 14 independent experiments; $n = 34$ (mOCs in contact with mOBs), $n = 67$ (mOCs not in such contact). Data are presented as means ± SDs. ****$p < 0.0001$ (Mann–Whitney test)

environments, enables the visualization of the local site of bone resorption in real time. To assess the bone-resorptive capacity of mOCs more precisely, we established a quantitative analysis method of bone-resorbing activity. Cell areas of mOCs were automatically extracted from the original maximum intensity projection (MIP) images (Fig. 2b) and mean pHocas-3 fluorescence intensity in the mOC area (pHocas-3 signal) and those outside of the mOC area (pHocas-3 noise) were measured. The bone resorbing index (BRI) was calculated as a signal-to-noise ratio (pHocas-3 signal/pHocas-3 noise) (Fig. 2c). We confirmed that the BRI of mOCs increased under osteoporotic conditions but decreased after bisphosphonate treatment, suggesting that the BRI quantitatively reflected the extent of bone-resorbing activity (which varied over time; Fig. 2d, e and Supplementary Fig. 2e–g). We also found that the BRIs of mOCs in contact with mOBs were significantly lower than those of mOCs lacking such contact (Fig. 2f–h and Supplementary Movie 2). In addition, we also analyzed the motility changes of mOCs in contact with mOBs. We have previously demonstrated that mOCs can be divided into two groups in terms of their motility and function, i.e., static resorbing osteoclasts (R state) and motile non-resorbing

osteoclasts (N state)[9]. Here we performed 3D colocalization analysis and quantification of the motility of mOCs, and concordantly with the previous study the motility of mOCs in contact with mOBs turned out to be significantly higher than those of mOCs lacking such contact (Supplementary Fig. 3a–g). These results also suggest that mOBs inhibit bone resorption of mOCs by direct cell–cell contact.

**Intermittent PTH treatment induces merged distribution.** Intermittent administration with PTH has been shown to induce osteo-anabolic action by modulating communication between osteoblasts and osteoclasts[15]; however, the mechanism underlying the action of PTH remains unclear. Here we examined the action of PTH on the communication of these two cell types using a murine model with intermittent treatments of teriparatide, a therapeutic form of PTH (1–34)[16]. After 3 weeks of PTH treatment, the murine model exhibited a significant increase in cortical bone volume, but not in cancellous bone volume, compared with untreated mice; significant increases in both cortical and cancellous bone volumes were observed after 6 weeks of treatment (Fig. 3a–c and Supplementary Fig. 4). On the other hand,

no significant increases in serum C-telopeptide of type 1 collagen (CTX) were found until 6 weeks after treatment (Supplementary Fig. 4). To analyze the time-course of the effect of PTH treatment, we compared the control conditions with 1, 3, and 6 weeks after PTH treatments (1w-PTH, 3w-PTH, and 6w-PTH). Upon PTH treatment, the number of mOCs first increased within 1 week (1w-PTH) (Fig. 3d, g); then, the number of mOBs with mixed distribution of mOBs and mOCs increased over 3 weeks after treatment (3w-PTH and 6w-PTH) (Fig. 3e, f, h). Magnified views of these images demonstrated a possible increase in direct cell–cell contact between mixed mOBs and mOCs (Fig. 3d–f). These results suggested that intermittent PTH treatment induced merged distributions of mOBs and mOCs with increased direct cell–cell contact between these two cell types.

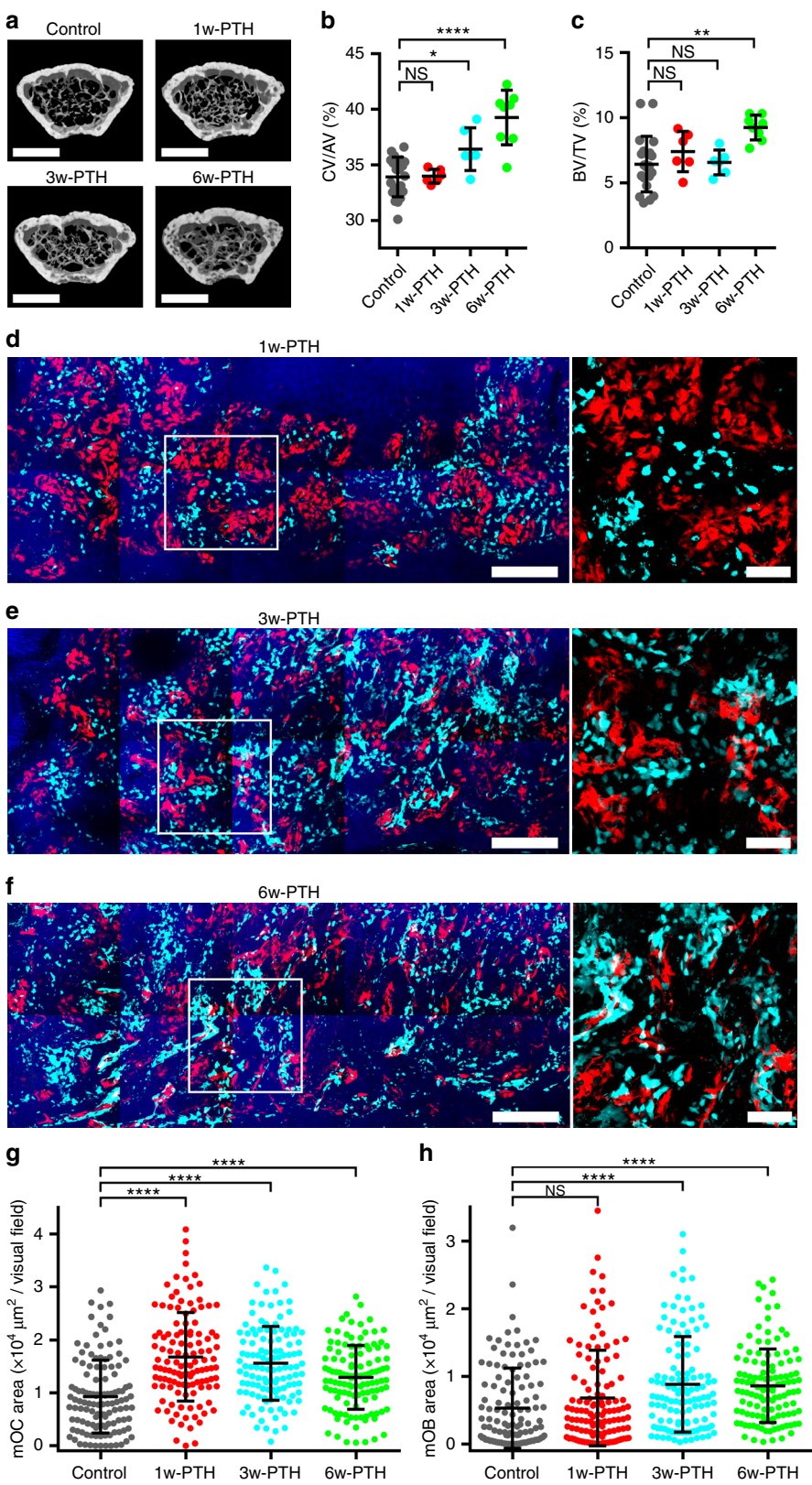

**Quantitative analyses of PTH-induced merged distribution.**
Tiling images showed that not only the number but also the distribution of mOBs and mOCs changed dramatically with the duration of intermittent PTH administration (Fig. 3d–h). To perform quantitative analyses of these changes, we developed a novel mathematical cell mixture analysis method, with hierarchical clustering and calculations of disparity for each cell type (Fig. 4a–e). The analysis was performed in four steps: (I) the cyan (mOBs) and red (mOCs) areas were binarized by Otsu's thresholding method (Fig. 4b, Step 1); (II) hierarchical clustering was performed based on the distance between each cluster, regardless of color (Fig. 4c, Step 2); (III) the color impurity of each cluster was calculated at each hierarchy (Fig. 4d, Step 3); and (IV) the cell mixture index (CMI) was calculated using a GLI test (Fig. 4e, Step 4). The Gini impurity is widely used in research areas of machine learning to measure the impurity of clusters or the goodness of prediction for a model. Similarly, in our study, to determine the number of clusters for a given threshold, impurity was defined as the weighted average of cyan/red area ratios of each cluster to measure the degree of mixture at that hierarchy. Calculating impurity for the set of clusters at each hierarchy from the top of the tree (all pixels were in one cluster), we obtained the characteristic curve (monotonously decreasing), which reached its minimum when all pixels in the cluster exhibited a single color. That is, the curve of a well-agglomerated pattern decreased rapidly with an increase in the number of clusters and the highly mixed pattern maintained a high value. Finally, the area under the curve was calculated as the index of mOB–mOC mixture distribution (CMI). The CMI value was normalized to be in the range of 0 to 1. CMIs were predicted to be lower if mOBs and mOCs formed mostly colonies of single cell types, whereas CMIs would be higher with high mOB–mOC mixture distribution (Fig. 4e). In addition, cell mixture analysis enabled measurement of the area of mOBs and mOCs in Step 1 at the same time.

Cell mixture analysis revealed that there were no significant differences in mOB–mOC mixture distribution between untreated and 1w-PTH-treated mice; however, both 3w- and 6w-PTH treatments led to significantly high mOB–mOC mixture distribution compared with untreated and 1w-PTH-treated mice (Fig. 4f). Taken together, the cell mixture analysis of large tile images demonstrated that intermittent PTH treatment significantly induced the merged distribution of mOBs–mOCs, possibly facilitating their cell–cell contact.

**Intermittent PTH treatment increases the number of contact.**
We next performed 3D colocalization analysis to explore the effects of intermittent PTH treatment. Regions of mOB–mOC contact were automatically detected at 30 min intervals over 8 h, and the contact number (events per hour) and contact durations were measured.

The results showed that 3w- and 6w-PTH treatment significantly increased the number of mOB–mOC contact compared with those of control and 1w-PTH treatment (Fig. 5a,

b and Supplementary Movies 3–6). The number of contact events normalized by surface areas of mOBs or mOCs were also increased under 3w- and 6w-PTH conditions (Fig. 5c, d), meaning that the number was elevated not simply by the increase in cell density but also by a genuine mechanism for promoting cell–cell contact driven by PTH. On the other hand, contact durations between mOB and mOC were unchanged irrespective of PTH treatment (Fig. 5e). Taken together, PTH treatment increased the number of the event of mOB–mOC contact, but did not affect the duration of each contact.

**Contact attenuates resorbing activity in PTH-treated bone.**
Under PTH treatment conditions, the number of mOCs was increased first, compared with those of mOBs (1w-PTH); in addition, most mOCs were free from mOBs and they secreted acids (Fig. 6a, b, e and Supplementary Movies 7, 8). In the later phases of PTH treatment, the number of mOBs increased with delayed proliferation. Moreover, they demonstrated increased contact with mOCs, which led to the inhibition of osteoclastic bone resorption (Fig. 6c–e and Supplementary Movies 9, 10).

We evaluated the bone-resorbing activity of mOCs for 4 h using the average value of all BRI values at 5 min intervals for 4 h (4h-BRI). We found that 4h-BRI values were dependent on the stage of PTH treatment, significantly correlating with the cell merge status, CMI (Spearman's correlation coefficient, $r = -0.5717$) (Fig. 6f). These results revealed the novel finding that mOBs attenuate bone-resorbing activity of mOCs via cell–cell contact, which is augmented by anabolic conditions in PTH-treated bone.

**Discussion**
The bone remodeling process takes place asynchronously at BMUs. In the bone remodeling process, old bone is resorbed by osteoclasts, followed by recruitment of osteoblast precursors that differentiate and replace the resorbed bone by osteoclasts. Therefore, it has been thought that bone resorption and formation do not occur simultaneously at the same BMU[1,2]. To date, communication between mOBs and mOCs has been questioned, and it has been generally difficult to detect direct binding between mOBs and mOCs in bone tissue sections based on conventional bone histomorphometric analyses. In this study, we applied intravital two-photon imaging to detect mOB–mOC contact in bone in vivo. Intravital two-photon bone imaging is advantageous, because it enables two-dimensional scanning in bone in a focal plane to observe cell shapes and the appearance of mOBs and mOCs in vivo. This approach is in marked contrast with conventional histomorphometric analyses by which the bone surface is represented as one-dimensional lines along the bone trabeculae. Although we are also able to perform in vitro experiments and observe the interaction between mOBs and mOCs, the in vitro scenario differed from that evident in vivo, because of the many limitations of in vitro experiments. First, we could not directly isolate mOBs and mOCs capable of performing

**Fig. 3** Intermittent PTH treatment induces merged distribution in vivo. **a–c** Micro-computed tomography (micro-CT) analysis of the distal, femoral metaphyseal regions. Twelve-week-old female mice were given the vehicle or PTH (40 µg kg$^{-1}$ per day, 5 days per week) via subcutaneous (s.c.) injection and were evaluated 1, 3, and 6 weeks later; $n = 20$ in the control group, $n = 6$–8 in each PTH-treated group. Data are presented as means ± SDs. *$p < 0.05$; **$p < 0.01$; ****$p < 0.0001$; NS, not significant (one-way ANOVA). **a** Representative micro-CT three-dimensional (3D) images. Scale bar, 1,000 µm. **b** Cortical bone ratios (cortical bone volume/total bone volume; CV/TV). **c** Bone matrix densities (bone volume/total volume, BV/TV). **d–f** Representative, intravital, two-photon, microscopy tiling MIP images of skull bone tissues of Col2.3-ECFP/TRAP-tdTomato mice taken 1, 3, or 6 weeks after PTH treatment (**d**, 1w-PTH; **e**, 3w-PTH; **f**, 6w-PTH). Cyan, mOBs expressing Col2.3-ECFP; red, mOCs expressing TRAP-tdTomato; blue, bone tissues (SHG). Scale bar, 300 µm. Magnified images of representative distributions of mOBs and mOCs in either group (right panels). Scale bar, 100 µm. **g**, **h** Areas of mOCs and mOBs per visual field; $n = 120$, collected from 12 tiling images from the six mice in each group. Data are presented as means ± SDs. ****$p < 0.0001$; NS, not significant (Kruskal–Wallis test)

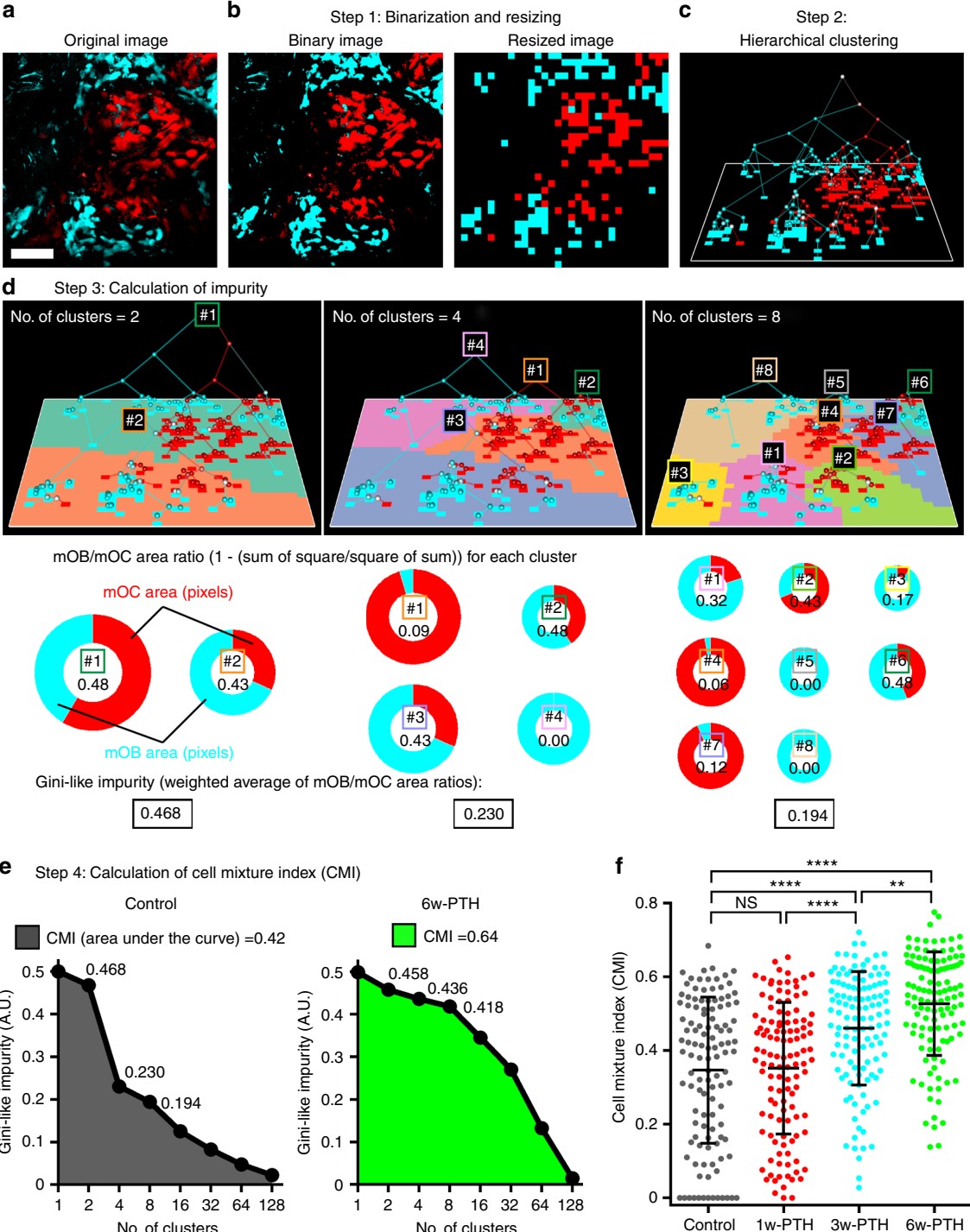

**Fig. 4** Quantitative analysis of PTH-induced merged distribution. **a–e** The procedure used for cell mixture analysis. **a** A representative, intravital, two-photon microscopy MIP image of the skull bone tissue of a Col2.3-ECFP/TRAP-tdTomato mouse held under control conditions. Cyan, mOBs expressing Col2.3-ECFP; red, mOCs expressing TRAP-tdTomato. Scale bar, 100 μm. **b** The areas of mOBs and mOCs shown in **a** were automatically binarized and the binarized image were resized (smaller). **c** Hierarchical clustering was performed based on the between-pixel differences, regardless of color. **d** To derive a threshold allowing the number of clusters to be determined, the Gini-like impurity (GLI) was calculated as the weighted averages of the mOB/mOC area ratios of each cluster at particular levels of the tree. As examples, we present the calculations for the cases in which the cluster numbers were 2, 4, and 8. The clusters used to calculate each threshold are shown in the same colors on the image. The ratios between the mOC and mOB areas are indicated by red and cyan in the circular charts; the areas of the circles reflect the areas occupied by the cells. **e** The GLI curves and the cell mixture index (CMI) values were calculated for the image shown in **4a** (left graph) and the magnified image shown in **3f** (right graph). The CMI was defined as the area under the GLI curve, which indicated the extent of mixing of the two types of cells within an image. **f** The CMI values per visual field in control mice and 1-w-, 3-w-, or 6-w-PTH-treated Col2.3-ECFP/TRAP-tdTomato mice; n = 120; collected from 12 tiling images from the six mice of each group. Data are presented as means ± SDs. **p < 0.01; ****p < 0.0001; NS, not significant (Kruskal–Wallis test)

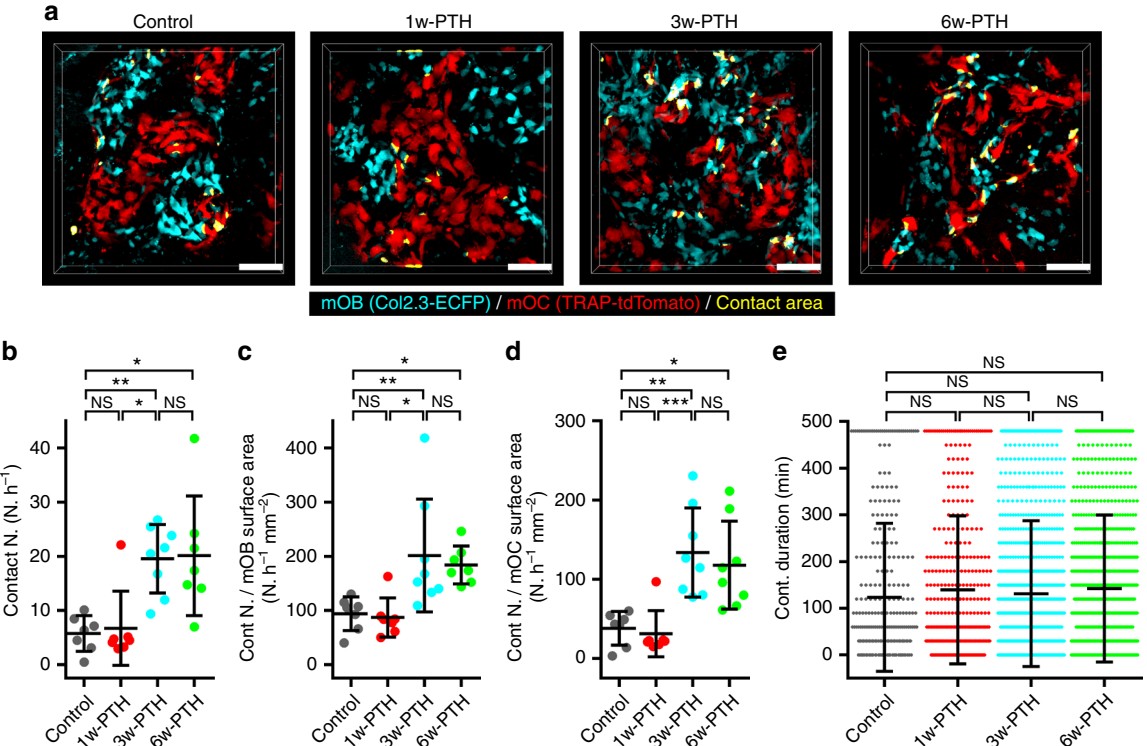

**Fig. 5** Intermittent PTH treatment increases the number of contact. **a** 3D colocalization analysis of representative images from control mice and 1-w-, 3-w-, and 6-w-PTH-treated Col2.3-ECFP/TRAP-tdTomato mice. Cyan, mOBs expressing Col2.3-ECFP; red, mOCs expressing TRAP-tdTomato. The contact areas were defined as the areas of colocalization of mOBs and mOCs, and are shown in yellow. Scale bar, 100 μm. **b** The number of mOB–mOC contacts. The duration of a mOB–mOC contact event was defined as the time from initial attachment to the end of mOB–mOC contact. **c** The numbers of contact events normalized by the surface areas of the mOBs. **d** The numbers of contact events normalized by the surface areas of the mOCs. **b**–**d** The data were collected from seven to eight independent experiments per group (control; $n = 7$, 1-w-PTH; $n = 7$, 3-w-PTH; $n = 8$, 6-w-PTH; $n = 7$). **e** The duration of mOB–mOC contact. Data were collected in seven to eight independent experiments performed per group (control; $n = 322$, 1-w-PTH; $n = 375$, 3-w-PTH; $n = 1,252$, 6-w-PTH; $n = 1,126$). Data are presented as means ± SDs. *$p < 0.05$; **$p < 0.01$; ***$p < 0.001$; NS, not significant (Kruskal–Wallis test)

time-lapse imaging from bones. Second, the physiological microenvironment of the in vivo bone marrow (e.g., the cell density and cytokine levels) are not reproduced in vitro. Thus, we chose to perform in vivo imaging.

In this study, we observed that mOBs and mOCs formed separate colonies under control conditions. This observation was consistent with the previous finding that the bone formation and bone resorption areas were separated from each other[1,2]. In addition, we found that the extent of direct contact of mOCs with mOBs was negatively correlated with the bone-resorbing activity of mOCs and positively correlated with the motility. Most mOCs in contact with mOBs displayed dendritic shapes with synapse-like projections toward mOBs and these projections moved actively while keeping contact with mOBs, leading to higher motility of mOCs. This finding is consistent with our previous report that mOCs undergo a transition between two functionally distinct states, i.e., static bone-resorptive 'R-type' and moving non-resorptive 'N-type'[9]; direct contact with mOBs would convert the R-type into N-type mOCs.

The bone remodeling process consists of distinct phases including activation, resorption, reversal, and formation. In the reversal phase, osteoclastic bone resorption ceases, which is accompanied by apoptosis of osteoclasts, whereas bone formation is activated[4]. The regions that exhibited mOB–mOC contact would be regarded as part of those in the reversal phase. The inhibitory signal from mOBs is suggested to be important for preventing the coincidence of bone resorption and formation in the same area, and for efficiently changing the phases from resorption to formation, which is reasonable for the bone remodeling process.

RANKL, an essential osteoclast differentiation factor expressed by cells of the osteoblast lineage, exists in both membrane-bound and soluble forms[17]. As cell–cell contact seems to be required for osteoclast formation during in vitro co-culture[18], it is believed that membrane-bound RANKL is the more potent form of the factor. Previous reports found that PTH significantly upregulated RANKL expression in late-stage (rather than early-stage) osteo-blasts growing in BMSC cultures[19]. In the present study, we showed that intermittent PTH treatment over 1 week significantly increased the number of mOCs without any increase in the number of mOBs or the extent of mOB–mOC contact. These results may suggest that direct cell–cell contact between mOBs and mOCs is not necessarily required for osteoclastogenesis, and that soluble RANKL secreted from mOBs may also potently induce mOCs in vivo, although we cannot completely exclude the possibility that mOBs in contact with early TRAP-negative osteoclast precursors can also promote osteoclastogenesis. It has also been reported that osteocytes (terminally differentiated osteoblasts) express RANKL[20,21] and regulate bone remodeling via PTH/PTH-related peptide type 1 receptor signaling[22,23]. Further studies are required to reveal the roles played by osteo-cytes; it is essential to visualize osteocytes in vivo.

In clinical studies, weekly injections of teriparatide increased bone mass and the levels of bone formation markers without affecting bone resorption[24]. However, the mechanism involved remains unclear. We found that intermittent PTH treatment for more than 3 weeks significantly increased bone volume without enhancing bone resorption, despite a significant increase in the number of mOCs. These phenomena may be attributable to

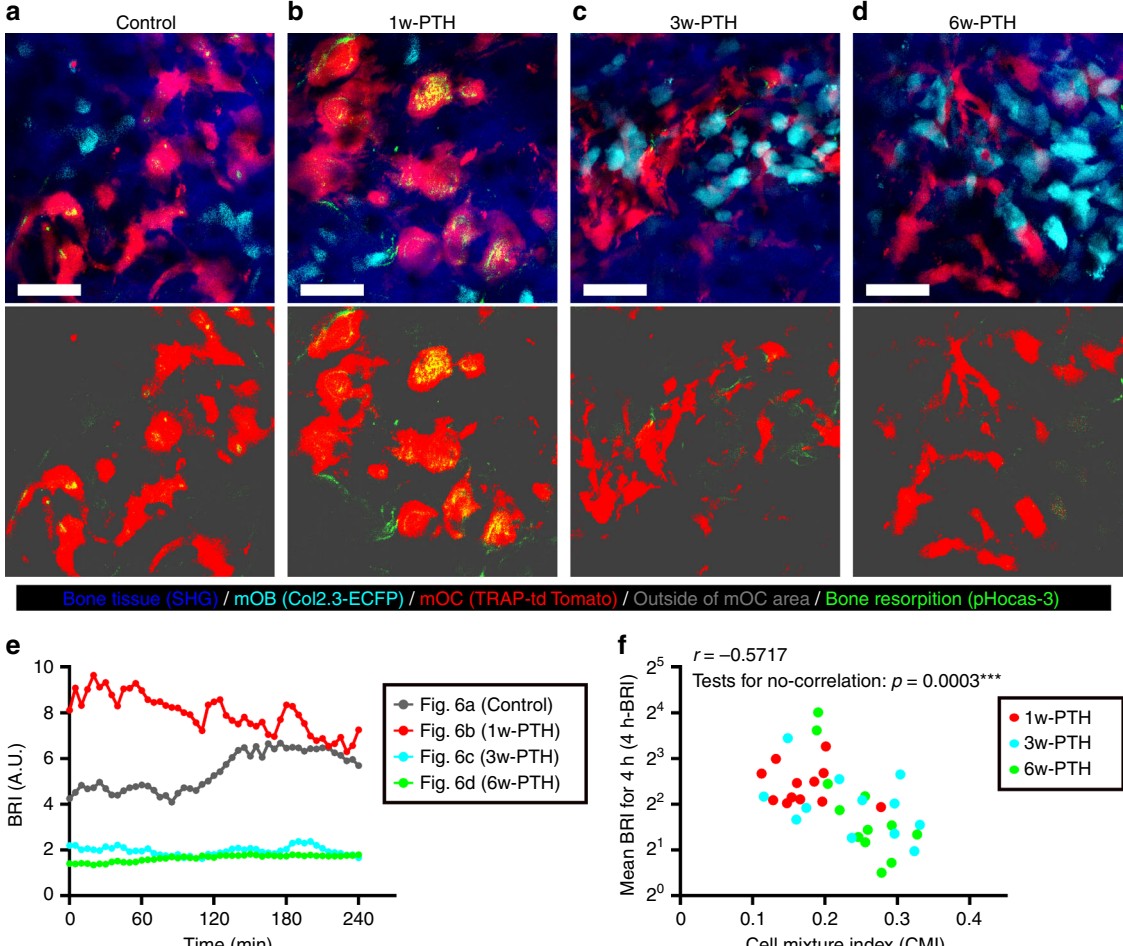

**Fig. 6** Direct contact attenuates resorbing activity in PTH-treated bone. **a–d** Representative MIP images of the bone-resorptive activities of control and 1-w-, 3-w-, and 6-w-PTH-treated Col2.3-ECFP/TRAP-tdTomato mice using a pH-sensing chemical probe, pHocas-3. The binarized mOC areas and the pHocas-3 signals are indicated in the bottom panels. Green, fluorescent signals from pHocas-3; Cyan, mOBs expressing Col2.3-ECFP; red, mOCs expressing TRAP-tdTomato; blue, bone tissues (SHG). Scale bar, 50 μm. **e** BRI time courses for the visual fields shown in **a–d**. **f** Correlations between CMI and the mean BRI values over 4 h; $n = 12$, collected from 8 to 10 mice per group. Data are presented as means ± SDs. ***$p < 0.001$ (Spearman's correlation coefficient, $r = -0.5717$; tests for no correlation, $p = 0.0003$.)

inhibitory signaling from mOBs to mOCs mediated via direct cell–cell contact.

Intravital two-photon microscopy in living bone has enabled detection of living bone cell types within bone tissues, allowing the analysis of the spatial-temporal relationship between mOBs and mOCs in vivo. Although the molecular mechanisms involved in direct cell contact remain elusive, this study clearly demonstrates an important concept that dynamic communication between mOBs and mOCs regulates bone homeostasis. These results may lead to the development of a new line of therapy for modifying the association properties of these two cell types.

## Methods

**Mice**. Female mice were used in all experiments. C57BL/6J mice were purchased from Crea Japan (Tokyo, Japan). The generation of the TRAP-tdTomato mice was described previously[9]. The Col2.3-ECFP mice were generated using a transgene expressing ECFP, driven by the 2.3 kb fragment of rat type I collagen α (1) promoter[11,12]. All mice were maintained under specific pathogen-free conditions and all animal studies were approved by the Institutional Animal Care and Use Committee of Osaka University.

**Intravital two-photon bone imaging**. Female Col2.3-ECFP, TRAP-tdTomato, and Col2.3-ECFP/TRAP-tdTomato mice (12–20 weeks of age) were used in all intravital two-photon bone imaging. All instruments for the intravital imaging technique were developed at the Center for Scientific Instrument Renovation and

Manufacturing Support (Osaka University). This novel stereotactic instrument consists of two parts made with stainless steel (SUS304). The upper part containing the head holder has one recess, in which the curvature radius is 28 mm. The center of the recess has a hexagonal window. A groove of 1.5 mm in width and 2 mm in depth adjoins the recess. The head holder is 3 mm thick and weighs 15 g (Supplementary Fig. 5a). The lower part containing the stage is composed of a 10 mm-thick metal plate; there are two cylinders with screw holes on the metal plate, which are used to fix the front teeth of the mouse on the metal plate (Supplementary Fig. 5b). The head holder can be fixed to the stage with two stainless screws (SUS304).

All surgical procedures were performed on mice subjected to isoflurane inhalation anesthesia. Before surgery, the mouse head was shaved and the skin was disinfected using 70% ethanol. An incision was made through the skin and the frontoparietal region of the skull bone was exposed. Marginal muscles and periosteum were resected, n-butyl cyanoacrylate glue (3 M Vetbond Tissue Adhesive, 3 M) was applied to the backside of the head holder, and ethyl-cyanoacrylate glue (Aron Alpha A Sankyo, Daiichi-Sankyo) was applied to the marginal area of the exposed bone, but not to the skull bone within the hexagonal window. Next, the head holder was placed on the same anatomical area of the skull bone using the skull bone suture as an anatomical landmark. The glue firmly secured the skull bone to the head holder (Supplementary Fig. 5c). After a few minutes, the head holder was fixed to the stage with two screws (Supplementary Fig. 5d). The continuous infusion line was positioned in the groove of the head holder and the recess of the head holder was kept fully loaded with phosphate-buffered saline (PBS) by an infusion syringe pump (KDS100, KD Scientific) (Supplementary Fig. 5e). Each mouse bearing the stereotactic instruments was placed in the imaging box. The frontoparietal regions of the skull bones were exposed and the internal surfaces of bones (thus those adjacent to the bone marrow cavities) were observed using two-photon excitation microscopy. During the

imaging experiments, the imaging box and the anesthetized mouse with the stereotactic instruments were kept warm at a constant temperature using a heat reserving plate and heated air. The mouse heart rate was monitored with an electrocardiogram monitor device (Nihon Kohden) and the anesthetic gas concentration was adjusted, with the heart rate as a guide.

**Settings of the two-photon microscopes.** The imaging systems consisted of a Nikon upright two-photon microscope (A1R-MP) equipped with an $\times 25$ water-immersion objective (APO, N.A. 1.1; Nikon) and a Carl Zeiss upright two-photon microscope (LSM 780 NLO) equipped with a $\times 20$ water-immersion objective (W Plan-Apochromat, N.A. 1.0). Both systems were laser-driven (Chameleon Vision II Ti:Sapphire; Coherent, Inc.).

Using a Nikon upright microscope, multi-fluorescent images were acquired by direct detection of fluorescence using four external non-descanned detectors equipped with dichroics and emission filters including an infrared-cut filter (DM685), three dichroic mirrors (DM458, DM506, and DM561), and four emission filters (417/60 for the second harmonic generation (SHG) image, 480/40 for ECFP, 534/30 for autofluorescence, and 612/69 for tdTomato). The excitation wavelengths were 860 nm for ECFP, and 900 nm for both of ECFP and tdTomato. Acquired images were subjected to channel unmixing using NIS Elements integrated software (Nikon) for autofluorescence and crosstalk reduction. After channel unmixing, constant $\gamma$ corrections were applied to all images (to enhance the signal-to-noise ratios) using NIS Elements integrated software: Cyan, $\gamma = 0.7$; tdTomato, $\gamma = 0.9$.

Using a Carl Zeiss upright microscope, spectral imaging was performed with specialized internal multi-photomultiplier detectors. Acquired raw images were subjected to spectral unmixing with ZEN software (Carl Zeiss) to create unmixed images that excluded autofluorescence. The excitation wavelength of 940 nm was used to simultaneously excite ECFP, tdTomato and pHocas-3. Intravital bone imaging experiments in the absence of pHocas-3 were performed using a Nikon two-photon microscope, whereas experiments in the presence of pHocas-3 were performed using a Zeiss two-photon microscope. After spectral unmixing, constant $\gamma$ corrections were applied to all images using NIS Elements integrated software to enhance the signal-to-noise ratio: tdTomato, $\gamma = 0.9$; and pHocas-3, $\gamma = 2.5$.

To obtain the tiling images, two sets of snapshot image stacks of 10 continuous visual fields were collected across the sagittal suture at a depth of 50–200 μm below the skull bone surface (5 μm vertical steps) with $\times 1.0$ zoom and with $512 \times 512$ $X$–$Y$ resolution. After channel unmixing, tiling images were stitched from MIP images of 10 continuous visual fields. For intravital time-lapse bone imaging of Col2.3-ECFP mice, image stacks were collected at 3 μm vertical steps at a depth of 50–150 μm below the skull bone surface with $\times 2.0$ zoom, $512 \times 512$ $X$–$Y$ resolution, and a time resolution of 2 min. For intravital time-lapse imaging of Col2.3-ECFP/TRAP-tdTomato mice using a Nikon two-photon microscope, 50 sequential image stacks were acquired at 1 μm vertical steps at a depth of 50–150 μm below the skull bone surface with $\times 1.0$ zoom, $512 \times 512$ $X$–$Y$ resolution, time resolution of 5 min to perform 3D-colocalization analysis, and 10 sequential image stacks were acquired at 3 μm vertical steps with $\times 2.0$ zoom, $512 \times 512$ $X$–$Y$ resolution, time resolution of 1 min to perform CDI analysis. For intravital time-lapse imaging of bone-resorptive activity in TRAP-tdTomato mice or Col2.3-ECFP/TRAP-tdTomato mice using pHocas-3, image stacks were collected at 3 μm vertical steps at a depth of 50–150 μm below the skull bone surface with $\times 2.0$ zoom, $512 \times 512$ $X$–$Y$ resolution, and time resolution of 5 min. The MIP images created for supplemental video were corrected for $XY$ drift using NIS Elements integrated software.

**Channel unmixing.** Channel unmixing was performed on all intravital bone imaging data from Col2.3-ECFP/TRAP-tdTomato mice using a Nikon two-photon microscope. The imaging data of the fluorescence spectra of ECFP, tdTomato, and autofluorescence were obtained using NIS Elements integrated software by manually selecting appropriate pixels on raw images; these spectral libraries were used for channel unmixing algorithms to create unmixed images in which each fluorescence was discriminated and autofluorescence was excluded (Supplementary Figs. 5f–h).

**Spectral unmixing.** Spectral unmixing was performed on intravital bone imaging data using a Zeiss two-photon microscope. Fluorescent spectra of SHG, auto-fluorescence, pHocas-3, ECFP, and tdTomato were obtained using the ZEN soft-ware by manually selecting appropriate pixels on true color images of intravital bone imaging of wild-type (WT) mice, WT mice administered with pHocas-3, Col2.3-ECFP mice, and TRAP-tdTomato mice, respectively. These spectral libraries were initially saved on a computer and used for spectral unmixing algorithms to create unmixed images in which each fluorescence was discriminated and autofluorescence was excluded (Supplementary Figs. 5i–k).

**Drug treatment.** Female C57BL/6J mice (12 weeks of age) and Col2.3-ECFP/TRAP-tdTomato mice (12–14 weeks of age) were given PTH (1–34) (teriparatide; Asahi Kasei Pharma Corporation) at 40 μg kg$^{-1}$ per day on 5 days per week via subcutaneous (s.c.) injection. Microstructure analysis, measurement of bone metabolic markers and intravital bone imaging were performed 1, 3, and 6 weeks after PTH treatment. PTH was not administered on the day of intravital imaging.

In the osteoporosis model, GST-RANKL (Oriental Yeast) (1 mg kg$^{-1}$ in PBS) was injected intraperitoneally into female TRAP-tdTomato mice 2 days before imaging. In bisphosphonate-treated animals, 100 μg kg$^{-1}$ risedronate (EA Pharma Co., Ltd) in PBS was intravenously injected 1 day before imaging.

**Cell deformation index analysis.** Assessment of the osteoclast motility was per-formed by using previously developed image analysis software CL-Quant 2.30 (Nikon) for tracking the morphological changes of osteoclasts[9]. Briefly, cell shapes were semi-automatically extracted by the software and the cell deformation index (CDI) was calculated as the ratio of the cell areas changed within 10 min to the total cell area at $t = 0$. High or low CDI value correlates with the high or low motility of mOCs (Supplementary Fig. 3e).

**Bone-resorbing index analysis.** A pH-sensing chemical probe (pHocas-3) dis-solved in PBS was injected s.c. at 5 mg per kg body weight daily into Col2.3-ECFP/TRAP-tdTomato mice, commencing 3 days before imaging[14]. We assessed the bone-resorbing ability of mOCs after image acquisition. mOC areas were binarized using Otsu's thresholding method and automatically extracted from the original MIP images (Fig. 2b). The mean pHocas-3 fluorescence intensities in mOC areas (pHocas-3 signals) and outside such areas (pHocas-3 noise) were measured. The BRI was the ratio of pHocas-3 signal to the pHocas-3 noise (Fig. 2c).

**Cell mixture analysis.** A novel mathematical method was developed to calculate the CMI, to indicate the degree of mOB–mOC mixture distribution. In addition, this cell mixture analysis enabled measurement of the cell areas of mOBs and mOCs at the same time by analyzing an MIP image of mOBs and mOCs visualized simultaneously by intravital two-photon microscopy. This method was imple-mented in R language (ver. 3.2.2), with the image processing aspect using the EBImage (ver. 4.10.1) package and the visualization component using the rgl (ver. 0.95.1367) package.

Step 1. Image binarization and resizing: As the first step in image analysis, both cyan and red cell areas were extracted from the input image (Fig. 4b). To automatically determine the threshold for binarization, we applied Otsu's method to a tiling image. Each threshold for binarizing cyan and red cells was calculated independently, and each binary image was processed using NIS Elements integrated software (Nikon, Japan) to exclude cyan areas $\leq 5$ μm$^2$ and red areas $\leq 10$ μm$^2$, except for those in mOBs and mOCs. Each processed image was divided into 10 MIP images (Fig. 4b). Cell area and CMI were calculated from each MIP image using NIS Elements integrated software (Nikon). When CMI was calculated from intravital time-lapse imaging data, the data were processed using NIS Elements to create MIP images of $X$–$Y$ planes drawn along the $Z$-axis and the time axis, and then cell areas were automatically extracted by Otsu's method. Resizing of the image was performed to reduce the computation time following analysis and enhancement of cellular patterns. The scale factor of resizing can be set arbitrarily (1/4 to 1/16), as the resulting CMI value was mostly not influenced by this size. In this study, we used 1/4 as a scaling factor for all CMI calculations, $512 \times 512$ to $128 \times 128$. It is noteworthy that a scaling factor of 1/16 was used only in Fig. 4, for ease of explanation.

Step 2. Hierarchical clustering: In the second step, we used hierarchical clustering in preparation for subsequent analysis to represent connectivity between cell areas. Euclidian distances were calculated between each pixel in the cell area extracted by Step 1 and Ward's method was used for the hierarchical clustering algorithm. At the initial stage of the process, each pixel was considered in a cluster of its own. The clusters were combined sequentially into larger clusters until all pixels converged into the same cluster. At each step, the clusters of shortest distance were combined (Fig. 4c). In this clustering step, the color of each pixel was not considered. Cutting this tree by thresholds, the tree was divided into clusters again.

Step 3. Calculation of impurity: Once the clusters were obtained for a given threshold, the impurity of the resulting clusters could be calculated. Impurity indicates how many pixels of different colors are merged into a cluster. In this study, we used a Gini-like impurity (GLI), which is used widely in machine learning[25] to measure the impurity of clusters or the goodness of prediction modeling. In this study, GLI was the weighted average of mOB/mOC area ratios ($I_{c_i}$), described as follows with respect to the set of clusters $C_m = \{c_1, \ldots, c_m\}$:

$$\mathrm{GLI}(C_m) = \sum_{i=1}^{m} \left( \frac{Y_{c_i} + R_{c_i}}{N} (I_{c_i}) \right) \qquad (1)$$

where $m$ is the number of clusters, $N$ is the total number of all pixels, and $Y_{c_i}$ and $R_{c_i}$ indicate the number of cyan pixels and red pixels in the $i$-th cluster $c_i$, respectively. $I_{c_i}$ represents the impurity of a certain cluster $c_i$, as given below:

$$I_{c_i} = 1 - \left( \frac{Y_{c_i}^2 + R_{c_i}^2}{\left(Y_{c_i} + R_{c_i}\right)^2} \right) \qquad (2)$$

The largest value of GLI is 0.5, when every cluster has the same amount of cyan and red pixels. In contrast, when each cluster consists of a single color, the value of

the impurity is 0. For example, as shown in the left panel of Fig. 4d (no. of clusters = 2), the two clusters consist of almost the same proportion of mOC and mOB areas, and the impurity values are high ($I_{c_1} = 0.48$, $I_{c_2} = 0.43$). Therefore, the overall impurity value for this separation is also high (GLI($C_{m=2}$) = 0.468). In contrast, in the case of the right panel of Fig. 4d (no. of clusters = 8), the overall impurity decreased (GLI($C_{m=8}$) = 0.194), because the clusters were becoming smaller, with each cluster containing only one kind of cell.

Step 4. Calculation of the CMI: The last step was CMI calculation. As mentioned previously, the process of decreasing impurity depends on its pattern (highly mixed or not). To quantify the distribution pattern of the two cell types, we defined CMI as the area under the curve of the GLI (Fig. 4e) in Eq. (3):

$$\text{CMI} = \frac{2}{\log_2(N+1)} \sum_{m=1}^{N} \left( \log_2(m+1) - \log_2 m \right) \text{GLI}(C_m)$$
$$= \frac{2}{\log_2(N+1)} \sum_{m=1}^{N} \log_2 \left( \frac{m+1}{m} \right) \text{GLI}(C_m) \qquad (3)$$

GLI($C_m$) is the impurity measure defined in Eq. (1) when the number of clusters is $m$. $N$ is the total number of cyan and red pixels. This equation indicates the log-weighted sum of impurity values from each set of clusters in the hierarchical clustering tree. Weight is calculated using the base 2 logarithm, because the structure of hierarchical clustering is a binary tree. In addition, the value of CMI is normalized to be in the range of 0 to 1. We introduced an approximation in Eq. (4) to reduce the calculation costs of Eq. (3):

$$\text{CMI} = \frac{2}{N'+1} \sum_{m'=\{2^0, 2^1, 2^2, 2^3, \ldots, 2^{N'}\}} \text{GLI}(C_{m'}) \qquad (4)$$

In this formula, we only calculated the impurity in the case in which the number of clusters can be expressed as an exponential of 2. $N'$ is the maximum number that satisfies $2^{N'} \leq N$. The resulting CMI value is only slightly affected by this approximation and it suppresses the computational cost drastically.

From the definition, impurity has the characteristics of monotonically decreasing with respect to the number of clusters. The patterns converge to zero impurity when the number of clusters equals the number of pixels. What matters is the timing of losing impurity. For a well-agglomerated pattern, impurity decreases rapidly with an increase in the number of clusters. Meanwhile, for a highly mixed pattern, high impurity levels are maintained despite smaller partitioning. Details of the method are also described in Supplementary Figs. 6–8 and Supplementary Discussion.

**3D colocalization analysis**. The number and duration of mOB–mOC contact was analyzed using Imaris software (Bitplane). A Sobel filter was used to detect cell edges in all depth slices, at all times, in all channels ($3 \times 3$, four directions for images acquired by a Nikon upright microscope with $\times 1.0$ zoom; $5 \times 5$, four directions for images acquired by a Nikon upright microscope with $\times 2.0$ zoom or a Carl Zeiss upright microscope). We next calculated the Lighten composites of the raw and edge images to yield edge-enhanced images. The surface tool of the Imaris software was used to perform automatic cell-surface segmentation of each cyan-positive and tdTomato-positive cell evident upon intravital time-lapse bone imaging. Cyan-stained surface objects $\leq 125\ \mu m^3$ in volume and tdTomato-stained surface objects $\leq 1,000\ \mu m^3$ in volume were not included in the analysis, as such groupings were unlikely to represent cells. The surface tool was then used to detect mOB–mOC contacts and to automatically create a new channel (yellow) revealing colocalized cyan and tdTomato voxels. The mOB–mOC contact time was that from the commencement of interaction to the end of mOB/mOC attachment. The Imaris tracking tool was used to measure contact numbers and durations.

**Microstructure analysis**. A cone-beam X-ray micro-computed tomography (CT) system (ScanXmate-RB090SS150; Comscantecno) was used to obtain CT images of isolated bone samples. The settings were as follows: tube voltage, 70 kV; tube current, 0.1 mA; and voxel size, 12.0 μm. The 3D images were reconstructed and analyzed with the aid of TRI/3D-BON software (RATOC System Engineering). Regions of interest were drawn 500 μm from the end of each epiphyseal growth plate to points 1.0 mm along the cortical wall.

**Measurement of bone metabolic markers**. All mice fasted for 12 h before blood collection via cardiac puncture 18 h after the last teriparatide dose was given and the samples were stored at −80 °C before analysis. The level of serum CTX, a bone resorption marker, was measured using a RatLaps enzyme immunoassay (Immunodiagnostic Systems) according to the manufacturer's instructions.

**Cell culture**. For in vitro osteoblast differentiation, primary bone marrow cells ($7.9 \times 10^5$ cells per cm²) from long bones of Col2.3-ECFP mice were suspended in culture medium (α-MEM containing penicillin, streptomycin, and 10% fetal bovine serum) for 2 days, to obtain BMSCs. The resultant BMSCs were cultured with osteogenic medium (50 μM ascorbic acid and 10 mM β-glycerophosphate)[26,27]. The medium was changed every 2 days. We evaluated the extent of ECFP fluorescence, ALP staining, and mineral formation (the latter was assessed via Alizarin

Red staining). ECFP fluorescence was observed using a Biostation IM-Q (Nikon), equipped with a BP434/17 excitation filter, DM452, and BP479/40 emission filter. For fluorescence-based observation for mineral formation, BMSCs were incubated with osteogenic medium supplemented with Alizarin Red S (0.25 mg ml⁻¹, Nacalai Tesque) for 30 min at 37 °C. Images were acquired using an A1 confocal microscope (Nikon).

**Quantitative real-time PCR**. Total RNA and complementary DNA were prepared using Maxwell 16 LEV Simply RNA Purification kits (Promega, WI) and Superscript III reverse transcriptase (Thermo Fisher Scientific), according to the manufacturer's protocol. Quantitative real-time PCR was performed using the Thermal Cycler Dice Real Time System TP800 (Takara, Japan). Expression of each sample was calculated relative to the β-actin housekeeping gene. The primer sequences were: *Alp*, 5′-CCCAAGGAAAAGAAGCACGTC-3′ and 5′-ACATTAGGCGCAG-GAAGGTCA-3′; *β-actin*, 5′-CTTCTACAATGAGCTGCGTG-3 and 5′-TCAT-GAGGTAGTCTGTCAGG-3′; *Bglap*, 5′-TGGCGACACTTACCGAGCTT-3′ and 5′-CCATGCCCCTTGTAGTAGCTGTA-3′; *Col1a1*, 5′-TAAGGGTCCC-CAATGGTGAGA-3′ and 5′-GGGTCCCTCGACTCCTACAT-3′; *ECFP*, 5′-ACG-TAAACGGCCACAAGTTC-3′ and 5′-AAGTCGTGCTTCATGTG-3′.

**Histological analysis**. Histological analysis in bone tissues was performed using Kawamoto's film method, according to the manufacturer's protocol[28]. Briefly, mice were perfused with 4% paraformaldehyde (PFA) with 20% sucrose for fixation and dissected bone tissues were further fixed with 4% PFA and 20% sucrose for 4 h at 4 °C. Samples were embedded in Super Cryoembedding medium (Section-LAB Co. Ltd) and frozen. Frozen samples were cut into 5 μm sections with a cryostat (Leica, CM3050). Sections were mounted with Fluoromount (Diagnostic BioSystems) or Vectashield Mounting Medium with 4′,6-diamidino-2-phenylindole (DAPI) (Vector Laboratories, Inc.). Images were acquired with an A1 confocal microscope (Nikon). Specific parameters (excitation laser wavelength, dichroic mirrors, and emission filters, respectively) were used for detecting fluorescence of DAPI (405 nm, DM495, and BP450/50), ELF97 ALP activity staining (405 nm, DM640, and BP540/30); ECFP (457 nm, DM515, and BP482/35); Alizarin Red staining (514 nm, DM640, and BP595/50); tdTomato and 7-Aminoactinomycin D (7AAD) (561 nm, DM640, and BP595/50) (Nikon). For nucleic acid staining (with the exception of DAPI), sections were incubated with 7AAD (1:25, eBioscience) for 45 min at 37 °C. Enzyme histochemistry fluorescence-based ELF97 ALP activity staining were performed, according to the manufacturer's protocol[29].

**Statistical analysis**. All data were analyzed using GraphPad Prism software (GraphPad Software, Inc.) and are presented as means ± SDs unless otherwise stated. Statistical analyses were performed with the aid of the two-tailed Student's t-test, Welch's t-test, or the Mann–Whitney test for between-group comparisons; one-way analysis of variance with Dunnet's multiple comparisons post-hoc test, or the Kruskal–Wallis test with Dunn's multiple comparisons post-hoc test was used for comparisons among three or more groups. Two-tailed Spearman's rank correlation coefficients were calculated to assess the relationships between pairs of variables. All data were checked for normality using both histograms and the Shapiro–Wilk test, with the aid of SPSS Statistics version 21 (IBM); they were checked for variation using the F-test or Bartlett's test. A $p$-value $< 0.05$ was considered to reflect statistical significance. All data are representative of those of at least three independent experiments unless otherwise indicated. We estimated the required sample sizes by considering variations and means, and sought to reach reliable conclusions using sample sizes that were as small as possible. Gender- and age-matched mice were randomly assigned to the groups of the in vivo experiments and no data point was excluded. The investigators were not blind during either the experiments or the outcome assessments.

**Data availability**. The data that support the findings of this study are available from the corresponding author upon reasonable request.

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

## Acknowledgements
This work was supported by CREST, Japan Science, and Technology Agency; Grants-in-Aid for Scientific Research (A) from the Japan Society for the Promotion of Science (JSPS to M.I.); Grant-in-Aid for Young Scientists (A) from JSPS (to J.K.); Grants-in-Aid for Young Scientists (B) from JSPS (to S.F.); Grant-in-Aid for Challenging Exploratory Research from JSPS (to H.M.); Grants-in-Aid for Scientific Research (C) from JSPS (to S.S.); grants from the Uehara Memorial Foundation (to M.I.); from the Kanae Foundation for the Promotion of Medical Sciences (to M.I.); and from the Takeda Science Foundation (to M.I. and S.F.).

## Author contributions
M.F. and M.I. conceived and designed the study. M.F. performed imaging experiments in assistance with J.K., M.S., M.U., H. Mizuno, Y. Iwamoto, A.M., and K.H. H. Maeda and K.K. provided a pH-sensing chemical fluorescent probe. S.S. and H. Matsuda contributed to the imaging data analysis. S.F., S.O., U.C., and A.L. contributed to the generation of reporter mice. T.I. and Y. Isogai contributed to the pharmacological analysis. M.K., T.K., and H.Y. discussed the experiments and results. M.F., J.K., and S.S. co-wrote the initial draft. M.I. revised the final draft.

## Additional information

**Competing interests:** T.I. and Y. Isogai are full-time employees of Asahi Kasei Pharma Corporation. The remaining authors declare no competing financial interests.

