## [Peer Review File · Nature Communications]

Editorial Note: This manuscript has been previously reviewed at another journal that is not operating a transparent peer review scheme. This document only contains reviewer comments and rebuttal letters for versions considered at Nature Communications. Mentions of prior referee reports have been redacted.

Reviewers' comments:

Reviewer #1 (Remarks to the Author):

This is an interesting manuscript from the laboratory of Dr. M. Ishii, a leader in the use of intravital two photon microscopy for imaging bone cells in the mouse calvarium. In this manuscript, D. Ishii and his colleagues developed a mouse model that expresses red and cyan fluorescent proteins in mature osteoclasts (mOCs) and osteoblasts (mOBs), respectively, to visualize these two cell populations simultaneously and their spatio-temporal relationship in vivo. Additionally, they were able to visualize bone resorbing activity by employing a pH-sensing fluorescent probe that is distinguishable from the two fluorescent proteins through spectral unmixing. Using these tools, they reported that bone resorption is confined to areas of bone covered by mOCs, and is furthermore restricted to mOCs not in contact with mOBs. While the frequency of contact between mOCs and mOBs is low at steady state, it is significantly increased after PTH treatment, as is their mixed distribution. These are novel and significant findings that will be of interest to a broad segment of scientific community. In my opinion, the manuscript is appropriate for publication in Nature Communications upon revision; however there are a number of concerns that need to be addressed first.

1. A statement is made in line 87 and again in line 135 that there exists a "repulsive property" between mOBs and mOCs. However there is no data supporting an active repulsion mechanism. Is it possible that they are simply attracted to different regions of the bone without actively repulsing each other?
2. The statement that contact with mOB inhibits mOC bone resorption is based on data shown in Figure 2. Therefore this is a critical figure; yet in Fig 2a (dotted line region) as well as Fig 2f and 2h (filled arrowheads), all designated as contact areas, there are visible gaps between mOBs and mOCs. The authors should provide a more concrete and precise definition of contact and details of how contact is analyzed.
3. Fig 2d and 2e are not sufficient to "confirm that BRI reflects quantitative bone-resorbing activity" (line 150). An independent control experiment is needed to demonstrate the relationship between BRI and bone-resorbing activity.
4. There seem to be a mistake in Figure 2 caption, lines 665-666. The mOC shown in f was not in contact while the mOC shown in e was in contact with mOB.)
5. It appears that there are a substantial number of cells on the endosteal surface that are neither mOBs nor mOCs (for example in Supple Fig 1j). Can the authors clarify what these negative cells are? Also in Fig. 4, the initial steps to binarize the images naturally exclude dim cells from the analysis. As the paper relies heavily on thresholding techniques such as Otsu's method, it will be important to discuss how their analysis could be sensitive to different choice of thresholds (i.e. exclusion of different levels of dim cells).
6. It will be helpful if the authors can point to what a basic multicellular unit (BMU) looks like in their two-photon images. Also please explain what is meant by "BMUs may influence adjacent BMUs" (line 266).
7. The authors previous reported that bone resorptive mOCs are static while non-resorptive mOCs are motile. In this work the authors show that bone-resorbing activity of mOCs is inhibited by contact with mOBs, implying that contact with mOBs should increase the motility of mOCs. At the

same time the authors also suggested that contact with mOBs could result in mOC apoptosis (reversal phase of bone resorption, lines 272-73). Can they provide time lapse imaging data to show whether contact with mOBs result in increased mOC motility or apoptosis?

8. In Fig. 5, how was it possible to determine contact frequency (per second) if time lapse images were acquired at 30 min intervals? Please clarify.

9. Suppl Fig 1 d & e image quality should be improved; also the caption of Suppl Fig 1e says nuclei (DAPI) is shown in blue but I do not see nuclei in the images.

10. It is unclear how frequently MIPs were used in the analysis. MIPs are appropriate for displaying 3D data in 2D but not for measuring cell-cell contact. Can the authors explain exactly how MIPs were used in the analysis? Overall, the authors need to clearly state when the images in the Figures are raw images or processed images (e.g. MIPs, thresholded, etc.)

Reviewer #2 (Remarks to the Author):

In their manuscript entitled "Dynamic modes of communication between mature osteoblasts and osteoclasts in vivo", Furuya et al use a dual reporter mouse and 2-photon imaging to analyze interactions between these two cell types at baseline and following intermittent PTH administration. This type of analysis has not been done before, and the authors calculate a new "cell mixing index" and utilize an osteoclast activity probe that they recently characterized to calculate a "bone resorbing index". They claim that PTH treatment increases the interactions between osteoclasts and osteoblasts (more mixing of cell types) and that these interactions inhibit bone resorption. These approaches provide new information that has not been provided by other types of analysis, and will be applicable to a wide variety of pathological settings. However, there are some concerns with the specifics of the methods used in these new approaches that should be addressed.

1. It is difficult to see many of the features in the videos that are mentioned in the text. It would be useful to add some annotation, at least in the first few frames to draw the eye to specifics. For example, in Video S1, arrowheads could indicate the synapse.

2. Figure 2 shows only 2 interactions between osteoclasts and osteoblasts. How many such interactions were observed and quantitated for the BRI. Also, it is not clear from the movie in S2 that this is really a separate osteoclast from its neighbor to the right. In many frames they seem to be connected.

3. In fig 3, how do you know that contacts are really lost, and not that they have gone out of the plane being visualized?

4. Most of the image analyses were done using hierarchical cluster structure of the image and measures the impurity (using an established measure, GLI) at different levels of the hierarchy. This makes the method sensitive to the cluster hierarchy. As it is presented, the clustering procedure uses the Euclidean distances between pixels in different clusters. When cells are close to each other (as they appear often in the examples), the order of clusters to be merged can be quite random depending on the execution order of the code and slight differences in pixel distances. As a result, the cluster hierarchy is unreliable at best, which in turn makes CMI unreliable. This issue could have contributed to the large variance in the plots of Fig 4(f).

The authors need to present careful validation of their methodology as well as comparison with more robust alternatives. For example, why not compute the GLI over a uniform partitioning of the image into rectangular regions? If a multi-scale measure is desired, one could consider varying the size of the rectangular regions, or using a quad-tree.

5. There are two other minor issues with the calculations of CMI. First, the current formulation (Eq. 3,4) penalizes images with smaller number of pixels. Ideally, the measure should be normalized in

someway by the total number of pixels (clusters). Second, the paper made the statement that "impurity has the characteristics of monotonically decreasing with respect to the number of clusters". This is not obvious in the context of this clustering-based measurement. Either a reference or a proof should be provided.

6. In Figs 3 and 4, $n=120$ from 6 mice per group for the image analysis. Please defend this as an appropriate statistical approach. We don't use each slice in a microCT as a separate data point, so why is that done for each field here? The fields should be added or averaged for each mouse to get back to $n=6$. This may mean that many of the indices will no longer be statistically significant. Alternatively, you should at least visualize the datapoints with each mouse by color or symbol so the reader can see the within and between animal variability. In this vein, the word "indicated" on line 174 is too strong for the data shown. "Suggested" would be more appropriate.

7. It is hard to see how Fig 5h is $n=6-8$. Please explain.

Several statements in the discussion go beyond what has been demonstrated:

Line 266 – there is no way to tell if the cells in contact are in the same or adjacent BMUs.

Line 267 – causality has not been proven, only correlation

Line 286 – what if the osteoblast contact is with early, TRAP-negative osteoclast precursors? These would not be visualized here

294 – the BRI was increased by a lot at 1 week, so how can you conclude that bone resorption was not increased. It is possible that some effects of PTH are local and thus not reflected in the serum CTX. The microCT only shows net bone, so increased resorption early in some compartments such as calvaria (which was not assessed in the microCT) could be balanced by formation. Bone formation has not been analyzed in these mice in situ.

Reviewer #3 (Remarks to the Author):

This manuscript describes studies carried out using a new technique to visualize living tissues. By using specific two-photon microscopy of transgenic mice the interactions of osteoclasts and osteoblasts could be followed in the mouse skull. Overall the manuscript is well written, the experiments appear well carried out and the data presented well. My only significant issue relates to the validation of the technique as it is important that the interpretation of the images is related to real cell interactions. Perhaps a simple ex vivo or in vitro experiment using the cells from the transgenic mice could easily demonstrate the accuracy of the imaging. Two-photon imaging could then be directly compared to normal imaging of the same cells at the same time. This would then validate the interpretation of the two-photon microscopy images. Overall the manuscript demonstrates an important step forward into our understanding of cell interactions.

Specific issues

1. Line 47 abstract – the terminology "segregated fashion" might be better explained.
2. Line 296 – The sentence beginning this line is difficult to understand.

Response to the Reviewers

We thank all the reviewers for their positive and constructive comments on our manuscript (no. NCOMMS-17-03885-T), “**Dynamic modes of communication between mature osteoblasts and osteoclasts in vivo**” by Furuya *et al.* In response to the comments, we performed additional experiments and revised the manuscript. We hope that we have responded appropriately and that the revised manuscript is now suitable for publication.

Response to Reviewer #1:

[Comment]

1. A statement is made in line 87 and again in line 135 that there exists a "repulsive property" between mOBs and mOCs. However there is no data supporting an active repulsion mechanism. Is it possible that they are simply attracted to different regions of the bone without actively repulsing each other?

[Response]

We agree; we now use the phrase “discrete property” rather than “repulsive property.”

“This simultaneous visualization revealed two in vivo modes of interaction in mOBs and mOCs: (i) a constitutive discrete property within a certain distance and (ii) direct cell-to-cell contact in a spatiotemporally limited fashion.” (page 5, line 2-4).

“These imaging results revealed at least two modes of communication between these two cell types: (i) discrete colony-forming exclusion with certain marginal zones, and (ii) spatiotemporally limited direct cell-to-cell contact.” (page 7, line 23 - page 8, line 1).

[Comment]

2. The statement that contact with mOB inhibits mOC bone resorption is based on data shown in Figure 2. Therefore this is a critical figure; yet in Fig 2a (dotted line region) as well as Fig 2f and 2h (filled arrowheads), all designated as contact areas, there are visible gaps between mOBs and mOCs. The authors should provide a more concrete and precise definition of contact and details of how contact is analyzed.

[Response]

We thank the reviewer for this instructive comment. We defined mOB–mOC contact as co-localization of cyan and red voxels. To precisely define the contact areas, we developed a 3D co-localization system as described in the original Figures 5a–c and the Methods section. Briefly, the original images were processed with a Sobel filter to

enhance the cell edges. Next, the surfaces of all cyan- and red-positive cells were segmented using the surface tool of Imaris software. Finally, areas of co-localization of the cyan and red voxels were automatically detected. A mOB–mOC contact was defined as commencing at the time of initial interaction and concluding at the end of mOB/mOC attachment.

We applied our 3D co-localization analysis for the original Figures 1 and 2, which revealed the precise areas of contact of mOBs and mOCs. We added these new data to the revised Figures 1h, 2f, and 2g; and Supplementary Figures 2a–d, and we have rewritten the text accordingly:

“We further investigated the extent of direct cell–cell contact between mOBs and mOCs, via quantitative three-dimensional (3D) co-localization analysis using a modified version of a previous method¹³. Briefly, the original images were processed with the aid of a Sobel filter to enhance the cell edges (Supplementary Figs. 2a, b). Then, the cell surfaces of all cyan-positive and red-positive cells were segmented with the aid of Imaris software (Supplementary Fig. 2c). Finally, areas of co-localization of the cyan and red color voxels were automatically detected (Supplementary Fig. 2d, and Fig. 1h).” (page 7, line 14-20).

We also repeated several experiments to confirm the BRIs of mOCs in contact with mOBs and of mOCs lacking such contact. The BRIs of mOCs in contact with mOBs were significantly lower than those of mOCs lacking such contact, suggesting that mOBs inhibit bone resorption by mOCs via direct cell–cell contact. We added these new data to the revised Figure 2h and have rewritten the text accordingly:

“We also found that the BRIs of mOCs in contact with mOBs were significantly lower than those of mOCs lacking such contact (Figs. 2f–h, and Supplementary Video 2). These results suggest that mOBs inhibit bone resorption of mOCs by direct cell–cell contact.” (page 8, line 16-19).

[Comment]

3. Fig 2d and 2e are not sufficient to "confirm that BRI reflects quantitative bone-resorbing activity" (line 150). An independent control experiment is needed to demonstrate the relationship between BRI and bone-resorbing activity.

[Response]

We have now performed imaging experiments using the pH probe under two extreme conditions to demonstrate the relationship between the BRI and bone-resorbing activity. One such condition was an osteoporotic scenario in which osteoclasts were aberrantly

activated. Mice were intraperitoneally injected with GST-RANKL (1 mg/kg) 2 days prior to imaging. The other condition was a scenario in which osteoclast functions were inhibited. Mice were intravenously injected with 100 µg/kg of risedronate, a bisphosphonate, 1 day prior to imaging. The BRI of mOCs increased under osteoporotic conditions, but fell after bisphosphonate treatment, suggesting that the BRI was, indeed, associated with bone-resorbing activity.

We added these new data to the revised Supplementary Figures 2e–g and have rewritten the text accordingly:

“We confirmed that the BRI of mOCs increased under osteoporotic conditions but decreased after bisphosphonate treatment, suggesting that the BRI quantitatively reflected the extent of bone-resorbing activity (which varied over time; Figs. 2d, e, and Supplementary Figs. 2e–g).” (page 8, line 13-16).

“In the osteoporosis model, GST-RANKL (Oriental Yeast) (1 mg/kg in PBS) was injected intraperitoneally into female TRAP-tdTomato mice 2 days prior to imaging. In bisphosphonate-treated animals, 100 µg/kg risedronate (EA Pharma Co., Ltd.) in PBS was intravenously injected 1 day before imaging.” (page 20, line 14-17).

[Comment]

4. There seem to be a mistake in Figure 2 caption, lines 665-666. The mOC shown in f was not in contact while the mOC shown in e was in contact with mOB.

[Response]

We apologize. As we explained in our response to comment 2, above, we had applied our 3D co-localization analysis for the original Figure 2 to identify the areas of contact between mOBs and mOCs more precisely. We have revised Figures 2d–g and have rewritten the Figure legends accordingly:

“(d, e) Images processed for BRI calculations (d for the mOC indicated with white asterisk in the outlined region of a, and e for the mOC indicated with black asterisk in the region delineated with a dotted line in a). (f, g) Magnified MIP images from the region outlined in a (f) and the region delineated by the dotted line in a (g), captured at 0, 160, and 320 min (upper panels). The 3D images yielded by co-localization analysis (bottom panels). The contact areas were those where mOBs and mOCs co-localized and are shown in yellow. The filled arrowheads show areas of mOB–mOC contact. The open arrowheads indicate separated mOBs and mOCs. The actual BRI values are shown to the right of the images.” (page 33, line 8-16).

[Comment]

5. It appears that there are a substantial number of cells on the endosteal surface that are neither mOBs nor mOCs (for example in Supple Fig 1j). Can the authors clarify what these negative cells are?

Also in Fig. 4, the initial steps to binarize the images naturally exclude dim cells from the analysis. As the paper relies heavily on thresholding techniques such as Otsu's method, it will be important to discuss how their analysis could be sensitive to different choice of thresholds (i.e. exclusion of different levels of dim cells).

[Response]

We thank the reviewer for this important comment. Indeed, substantial numbers of cells on the endosteal surface are neither mOBs nor mOCs. We previously showed that monocytes (including osteoclast precursors) may be found on the endosteal surface (Ishii et al., *Nature*, 2009; Ishii et al., *J Exp Med*, 2010). Another report found that pre-osteoblasts could be found on the endosteal surface (Nakashima et al., *Cell*, 2002). In the present study, we focused on communication between mature osteoclasts and osteoblasts, and we suggest that it would be beyond the scope of our paper to discuss the “dim” cells in more detail.

However, as the reviewer correctly pointed out, mOC and mOB area measurements are drastically affected by whether low-intensity areas are considered to be cell areas. However, it is very difficult to distinguish cellular from non-cellular regions via intravital imaging; no algorithm discriminating dim cells (or even negative cells) from “bright” cells has yet been developed. The use of “experience” to determine a threshold value is not appropriate; this is subjective. We used an automatic thresholding method to ensure (at least) reproducibility; we processed all images in the same way. In particular, Otsu's method yielded results that were similar to those that would have been obtained had we used our “experience” alone.

The following (reviewer-only) Figure A shows how CMIs were calculated after binarizing upon application of various thresholding methods to the control and 6w-PTH images. The typical patterns for each example are shown in (a). Images binarized using Otsu's method are shown in (b), whereas those obtained by employing Kittler's method (which we did not use but which is well-known) are shown in (c). Figures (d) to (g) were obtained by binarization using the percentile values of the intensity histograms as thresholds. When the 50-percentile values (the median intensity distributions) were used as thresholds, the red and cyan pixels each occupied half the entire image (the overlapping area is shown in white). When the 95-percentile values were used, 95% of the image areas were judged to be background. Figure (h) shows the CMI values obtained when the various thresholding methods were applied to the control and 6-w-PTH images. The

Figure shows that the relationships among the CMI magnitudes did not vary although the CMI values did, in fact, vary depending on the thresholding method used (the CMI of the 6-w-PTH image was higher than that of the control). Thus, the mOCs and mOBs are well-mixed in the 6-w-PTH image.

The Figure shows that when a global thresholding method is used, the CMI value indicating the extent of mixing is robust at least at the level of the ordinal scale. Such robustness is afforded by the use of a clustering algorithm when calculating the CMI. Generally, obvious high-intensity cellular areas tend to be located at the centers of the lumps, and they tend to be surrounded by ambiguous low-intensity regions. The distinct, high-intensity cell areas also form the cores of the clustering trees. Thus, the “thresholding problem” (whether the low-intensity regions are “dim cells” or background) is reduced.

(a) original

(b) Otsu

(c) Kittler

(d) 50-percentile

(e) 75-percentile

(f) 90-percentile

(g) 95-percentile

binarized mOB area / binarized mOC area / both areas overlapped

Reviewer-only Figure A

[Comment]

6. It will be helpful if the authors can point to what a basic multicellular unit (BMU) looks like in their two-photon images. Also please explain what is meant by "BMUs may influence adjacent BMUs" (line 266).

[Response]

We considered a colony composed of only mOBs or mOCs to constitute a BMU and assumed that a BMU might influence adjacent BMUs via mOB-mOC contact; however, we have no evidence to support this idea. As BMUs are irrelevant to our thesis, we removed the following text:

"In this study, we observed that mOBs and mOCs formed separate colonies under control conditions. This observation was consistent with the previous finding that the bone formation and bone resorption areas were separated from each other^{1,2}." (page 13, line 15-17).

[Comment]

7. The authors previous reported that bone resorptive mOCs are static while non-resorptive mOCs are motile. In this work the authors show that bone-resorbing activity of mOCs is inhibited by contact with mOBs, implying that contact with mOBs should increase the motility of mOCs. At the same time the authors also suggested that contact with mOBs could result in mOC apoptosis (reversal phase of bone resorption, lines 272-73). Can they provide time lapse imaging data to show whether contact with mOBs result in increased mOC motility or apoptosis?

[Response]

We performed time-lapse imaging for an extended period (4–8 h) to observe the communication between mOCs and mOBs. The experimental conditions used in the current study differed from those of our previous work. We earlier explored rapid dynamic changes in mOCs (of the static R type or the motile N type) over 20–30 min (Kikuta et al., *J Clin Invest*, 2013). As the time courses of the two phenomena (the rapid R/N conversion of mOCs and the slow interaction between mOCs and mOBs) differed, we could not determine whether non-resorptive mOCs in contact with mOBs were moving. Likewise, we could not detect mOC apoptosis using our current imaging protocol; this is a limitation of our study. In response to the reviewer's concern, we have added the following new text:

"Nevertheless, because the time courses of the two phenomena (the rapid R/N conversion of mOCs and the slow interaction between mOCs and mOBs) differed, we could not determine whether non-resorptive mOCs in contact with mOBs were moving."

(page 13, line 22 - page 14, line 1).

[Comment]

8. In Fig. 5, how was it possible to determine contact frequency (per second) if time lapse images were acquired at 30 min intervals? Please clarify.

[Response]

We agree. We now use the phrase “contact number” or “contact number per hour” in the revised manuscript and Figures 5b–d.

[Comment]

9. Suppl Fig 1 d & e image quality should be improved; also the caption of Suppl Fig 1e says nuclei (DAPI) is shown in blue but I do not see nuclei in the images.

[Response]

We agree that the original images were not very good. We have replaced the images in Supplementary Figures 1d and e with images of higher resolution. We have also deleted “DAPI” from the legend of Supplementary Fig. 1e.

[Comment]

10. It is unclear how frequently MIPs were used in the analysis. MIPs are appropriate for displaying 3D data in 2D but not for measuring cell-cell contact. Can the authors explain exactly how MIPs were used in the analysis? Overall, the authors need to clearly state when the images in the Figures are raw images or processed images (e.g. MIPs, thresholded, etc.)

[Response]

We thank the reviewer for this important comment. We performed all contact analyses using 3D imaging data. MIP imaging data can be employed to localize cells and probes. We used MIP data to derive CMI and BRI values.

CMI analysis was employed to evaluate the extent of mOB–mOC mixing. Mature OBs and OCs are localized along endosteal surfaces. Although the endosteal surfaces of skull bones exhibit some concavo–convex areas, most are nearly flat or have gentle slopes (Supplementary Figure 1h). The use of MIP to display 3D data in 2D is appropriate when calculating CMI.

We used BRI analysis to quantitatively assess the bone-resorptive capacity of mOCs. Osteoclasts attached to bone surfaces resorb bones by secreting acid and enzymes into the resorption lacunae. Thus, pH probe signaling was evident between the surface of the bone and the osteoclast basal membrane (Maeda, et al., *Nat Chem Biol*, 2016). We

estimated the fluorescence intensity of the pH probe associated with each mOC by extracting the pH probe signal from mOC area evident in the MIP images. The use of MIP to display 3D data in 2D is appropriate when calculating BRI.

We added information to the relevant Figure legends regarding why we used MIP image data.

Response to Reviewer #2:

[Comment]

1. It is difficult to see many of the features in the videos that are mentioned in the text. It would be useful to add some annotation, at least in the first few frames to draw the eye to specifics. For example, in Video S1, arrowheads could indicate the synapse

[Response]

We thank the reviewer for this instructive comment. We agree; we added arrowheads and a brief explanation to the revised Video S1.

[Comment]

2. Figure 2 shows only 2 interactions between osteoclasts and osteoblasts. How many such interactions were observed and quantitated for the BRI. Also, it is not clear from the movie in S2 that this is really a separate osteoclast from its neighbor to the right. In many frames they seem to be connected.

[Response]

We performed 14 independent imaging experiments and 3D co-localization analyses and calculated the BRIs of 34 mOCs in contact with mOBs and 67 mOCs lacking such contact. The former BRIs were significantly lower than the latter, suggesting that mOBs inhibit bone resorption by mOCs via direct cell–cell contact. We added these data to the revised Figure 2h, and we have rewritten the text accordingly:

“We also found that the BRIs of mOCs in contact with mOBs were significantly lower than those of mOCs lacking such contact (Figs. 2f–h, and Supplementary Video 2). These results suggest that mOBs inhibit bone resorption of mOCs by direct cell–cell contact.” (page 8, line 16-19).

*“(h) BRI of mOCs in contact, or not, with mOBs. Images were collected from 14 independent experiments; n = 34 (mOCs in contact with mOBs), n = 67 (mOCs not in such contact). Data are presented as means ± SDs. ****p < 0.0001 (Mann–Whitney test).”* (page 33, line 16-19).

In response to the reviewer's comment, we performed 3D co-localization analysis in the original Figure 2 to define the areas of contact between mOBs and mOCs more precisely. We replaced Video S2 with a clearer video.

[Comment]

3. In fig 2, how do you know that contacts are really lost, and not that they have gone out of the plane being visualized?

[Response]

We thank the reviewer for this important comment. We defined "mOB–mOC contact" as the area of co-localization of cyan and red voxels. To define contact areas precisely, we developed a 3D co-localization system as described in the original Figures 5a–c and the Methods section. Briefly, the original images were processed with a Sobel filter to enhance the cell edges. Next, the surface of each cyan- and red-positive cell was segmented using the surface tool of Imaris software. Finally, areas of co-localization of cyan and red voxels were automatically detected. A mOB–mOC contact was defined as commencing at the time of initial interaction and concluding at the end of mOB/mOC attachment.

We applied our 3D co-localization analysis for the original Figure 2, which lists the precise contact areas between mOBs and mOCs. We added these new data to the revised Figures 2f, and 2g; and Supplementary Figures 2a–d, and we have rewritten the text accordingly:

"We further investigated the extent of direct cell–cell contact between mOBs and mOCs, via quantitative three-dimensional (3D) co-localization analysis using a modified version of a previous method¹³. Briefly, the original images were processed with the aid of a Sobel filter to enhance the cell edges (Supplementary Figs. 2a, b). Then, the cell surfaces of all cyan-positive and red-positive cells were segmented with the aid of Imaris software (Supplementary Fig. 2c). Finally, areas of co-localization of the cyan and red color voxels were automatically detected (Supplementary Fig. 2d, and Fig. 1h)." (page 7, line 14-20).

As the reviewer correctly pointed out, we cannot completely exclude the possibility that some contacts may exit the visual field during intravital time-lapse imaging, although we would emphasize that all image stacks were collected at large z-widths and that almost all mOB/mOC interactions were observed in their entirety. Nevertheless, in response to the reviewer's concern, we now mention this limitation of our intravital imaging analysis:

“This allowed us to estimate the extent of direct cell–cell contact between mOBs and mOCs, with the limitation that some contacts may have exited the visual field during intravital time-lapse imaging.” (page 7, line 20-22).

[Comment]

4. Most of the image analyses were done using hierarchical cluster structure of the image and measures the impurity (using an established measure, GLI) at different levels of the hierarchy. This makes the method sensitive to the cluster hierarchy. As it is presented, the clustering procedure uses the Euclidean distances between pixels in different clusters. When cells are close to each other (as they appear often in the examples), the order of clusters to be merged can be quite random depending on the execution order of the code and slight differences in pixel distances. As a result, the cluster hierarchy is unreliable at best, which in turn makes CMI unreliable. This issue could have contributed to the large variance in the plots of Fig 4(f).

The authors need to present careful validation of their methodology as well as comparison with more robust alternatives. For example, why not compute the GLI over a uniform partitioning of the image into rectangular regions? If a multi-scale measure is desired, one could consider varying the size of the rectangular regions, or using a quad-tree.

[Response]

Clustering methods search for data patterns by joining or dividing elements. Various clustering methods have been proposed, and all have both advantages and disadvantages. No optimal method is obvious. We used hierarchical clustering because this is the simplest method, as relatively few parameters must be defined in advance. We employed the complete linkage algorithm (one of several hierarchical clustering algorithms) because computation is fast and the resulting clusters tend to be compact. Complete linkage clustering avoids one drawback of the single linkage method (the so-called chaining phenomenon) wherein clusters formed via single linkage clustering may be forced together because single elements are close, even though many elements of each cluster may be very distant. On the other hand, as the reviewer commented, the non-uniqueness of the joining order may have created an unstable hierarchical structure if it had been possible that one cluster had multiple equal-nearest neighbors. Additionally, the complete linkage algorithm overemphasizes the importance of outliers, which can create large changes in the hierarchical structure. Next, we should note the advantages and disadvantages of the uniform partitioning method mentioned by the reviewer. In contrast

to the hierarchical clustering that we employed, uniform partitioning is a divisive clustering method. An image is divided into uniform rectangular regions in a stepwise manner. Certainly, the clustering afforded by uniform partitioning may be robust from the viewpoint of the processing order. However, uniform partitioning is associated with a different problem. When the image center moves even slightly, the resulting clusters change drastically. In other words, uniform partitioning is vulnerable to shift and drift.

Below, we present some patterns exhibiting small perturbations of the original images. We used several hierarchical clustering algorithms (Ward's method, complete linkage, single linkage, average linkage, McQuitty's method, median linkage, and centroid linkage), and uniform partitioning, to explore differences in the CMIs generated. Reviewer-only Figure B (a) shows the typical patterns of control and PTH 6-week images, respectively. Reviewer-only Figure (b) contains examples of perturbed images (rotated, shifted, or after the addition of salt-and-pepper noise). Reviewer-only Figure (c) shows the CMI values, calculated using various algorithms, of the perturbed images.

Ward's method was the most robust [the leftmost column of (c)]. This algorithm is widely used in the field of data mining. The computation time is slightly longer than that required for complete linkage, but it generates well-balanced clusters even when outliers are present. The complete linkage algorithm [the second column from the left of (c)] exhibits dispersions when a pattern is rotated. This reflects the problem that arises when the joining order of nearby cell areas varies, as the reviewer (correctly) states. The CMIs calculated using simple uniform partitioning (the rightmost column) fluctuate, especially under rotation and shifting. The shifting susceptibility is caused by the fact that the image is divided into halves without reference to the pattern, and the structure changes drastically if the center of the image is deviated. The rotation susceptibility is caused by a change in the order of division (i.e., up-down first or left-right first). As the reviewer mentioned, a Quad-tree algorithm might be applicable and would be expected to be robust against rotation. However, the Quad-tree algorithm suffers from the same shift problem: a small image shift causes a large change in the cluster structure. Of course, the use of a more complicated algorithm (such as nonuniform quad partitioning or an analog thereof) might improve robustness in terms of shifting. However, this is a difficult problem. The ultimate clustering algorithm must be position-, rotation-, scale-, and initial value-invariant. We very much appreciate the reviewer's comments, and we will seek to develop such a method in the future.

Based on the above, we now use Ward's hierarchical clustering method, which is the most robust. All original data (including those in Figure 4) have been replaced by data produced by application of Ward's method. Although the computational times increased,

we consider variations caused by the algorithm (i.e., artifacts, in a sense) to have been suppressed and believe that the results are appropriate in a biological context. We added the following new text:

“Euclidian distances were calculated between each pixel in the cell area extracted by Step 1, and Ward’s method was used for the hierarchical clustering algorithm.” (page 22, line 8-10).

binarized mOB area / binarized mOC area

(b)

Reviewer-only Figure B

[Comment]

5. There are two other minor issues with the calculations of CMI. First, the current formulation (Eq. 3,4) penalizes images with smaller number of pixels. Ideally, the measure should be normalized in some way by the total number of pixels (clusters). Second, the paper made the statement that "impurity has the characteristics of monotonically decreasing with respect to the number of clusters". This is not obvious in the context of this clustering-based measurement. Either a reference or a proof should be provided.

[Response]

Because of our error (for which we apologize), we revised equations 3 and 4 as follows:

Before:

$$CMI = \frac{2}{\log_2 N} \sum_{m=1}^N \left(\log_2 \left(\frac{m}{m-1} \right) \right) GLI(C_m) \quad (3)$$

$$CMI = \frac{2}{N'+1} \sum_{m'=\{2^0, 2^1, 2^2, 2^3, \dots, 2^{N'}\}} GLI(C_{m'}) \quad (4)$$

After:

$$\begin{aligned} CMI &= \frac{2}{\log_2(N+1)} \sum_{m=1}^N (\log_2(m+1) - \log_2 m) GLI(C_m) \\ &= \frac{2}{\log_2(N+1)} \sum_{m=1}^N \log_2 \left(\frac{m+1}{m} \right) GLI(C_m) \end{aligned} \quad (3)$$

$$CMI = \frac{2}{N'+1} \sum_{m'=\{2^0, 2^1, 2^2, 2^3, \dots, 2^{N'}\}} GLI(C_{m'}) \quad (4)$$

We would like to explain the meanings of the equations using the following reviewer-only Figure C. Figure (a) shows a binarized image (512x512 pixels) and (b) shows a resized image (32x32 pixels) of (a). In Figure (b), the total number of red and cyan pixels is 133, i.e., N=133. A bar plot of the impurity values for every cluster set is shown in Figure (c) (constructed without logarithmic weighting of the cluster numbers). In such a case, if we were to calculate a "CMI" value in a manner similar to the way in which CMIs are usually calculated, the value would be the sum of the bars divided by 133 and multiplied by 2 (to normalize 0 to 1). This seems to underestimate the extent of mixture distribution. It is inappropriate to treat large and small changes equally; this is associated with a risk of miscalculation of cell areas. Therefore, we used the weighted averages (with respect to the cluster numbers) as shown in equation (3) and Figure (d). Finally, we

introduced an approximation [equation (4) and Figure (e)] to reduce computational costs, as described in the manuscript. In Figure (e), the number of impurity calculations is drastically reduced, from $N=133$ to $N'=8$, without the loss of any quantification power (N' is the maximum number satisfying: $2^{N'} \leq N$, i.e., $2^8 = 128 \leq 133$).

Reviewer-only Figure C

The fact that the GLI decreases monotonously with an increasing number of clusters is proven below (via a contradiction approach). A prerequisite is the formation of a hierarchical clustering tree. Slicing of the tree using defined thresholds determines the cluster number. To increase the number of clusters from m to $m+1$, a certain cluster is subdivided into two clusters. Let d be the cluster that will be divided at step C_m and d_1 and d_2 the clusters thus obtained prior to step C_{m+1} . At this time, the other clusters are not at all affected, and neither the impurity nor weight of any cluster used to calculate the GLI will change in a cluster other than d (as shown in the following reviewer-only Figure D):

Thus, if the GLI does not decrease monotonously, there exists a situation that satisfies $GLI(C_m) < GLI(C_{m+1})$. From equation (1):

$$GLI(C_m) = \sum_{i=1}^m \left(\frac{Y_{c_i} + R_{c_i}}{N} (I_{c_i}) \right) \quad (1)$$

considering that $GLI(C_m) < GLI(C_{m+1})$ is equivalent to a comparison between the weighted impurity of cluster d at C_m and those of d_1 and d_2 at $C_m + 1$; that is,

$$\frac{Y_{c_d} + R_{c_d}}{N} (I_d) < \frac{Y_{c_{d_1}} + R_{c_{d_1}}}{N} (I_{c_{d_1}}) + \frac{Y_{c_{d_2}} + R_{c_{d_2}}}{N} (I_{c_{d_2}}) \quad (i).$$

Now, cluster d_1 and cluster d_2 are subsets of cluster d , and the following equations (ii, iii) are always true:

$$Y_{c_d} = Y_{c_{d_1}} + Y_{c_{d_2}} \quad (ii)$$

$$R_{c_d} = R_{c_{d_1}} + R_{c_{d_2}} \quad (iii).$$

Using

$$I_{c_i} = 1 - \left(\frac{Y_{c_i}^2 + R_{c_i}^2}{(Y_{c_i} + R_{c_i})^2} \right) \quad (2),$$

as the definition of impurity, we can transform inequality (i) into inequality (iv) in the following way:

$$\frac{Y_{c_d} + R_{c_d}}{N} \left(\frac{2Y_{c_d}R_{c_d}}{(Y_{c_d} + R_{c_d})^2} \right) < \frac{Y_{c_{d_1}} + R_{c_{d_1}}}{N} \left(\frac{2Y_{c_{d_1}}R_{c_{d_1}}}{(Y_{c_{d_1}} + R_{c_{d_1}})^2} \right) + \frac{Y_{c_{d_2}} + R_{c_{d_2}}}{N} \left(\frac{2Y_{c_{d_2}}R_{c_{d_2}}}{(Y_{c_{d_2}} + R_{c_{d_2}})^2} \right) \quad (iv).$$

Now, divide by N and move the terms on the right to the left as follows:

$$\frac{2Y_{c_d}R_{c_d}}{(Y_{c_d} + R_{c_d})} - \frac{2Y_{c_{d_1}}R_{c_{d_1}}}{Y_{c_{d_1}} + R_{c_{d_1}}} - \frac{2Y_{c_{d_2}}R_{c_{d_2}}}{Y_{c_{d_2}} + R_{c_{d_2}}} < 0. \quad (v).$$

Substitute equations (ii) and (iii) into inequality (v) as follows:

$$\frac{2Y_{c_d}R_{c_d}}{(Y_{c_d} + R_{c_d})} - \frac{2Y_{c_{d_1}}R_{c_{d_1}}}{Y_{c_{d_1}} + R_{c_{d_1}}} - \frac{(Y_{c_d} - Y_{c_{d_1}})(R_{c_d} - R_{c_{d_1}})}{(Y_{c_d} + R_{c_d}) - (Y_{c_{d_1}} + R_{c_{d_1}})} < 0. \quad (vi).$$

Finally, after ensuring that all fractions have common denominators, we obtain the following inequality:

$$(Y_{c_d} + R_{c_d})(Y_{c_{d_1}} + R_{c_{d_1}}) \left((Y_{c_d} + R_{c_d}) - (Y_{c_{d_1}} + R_{c_{d_1}}) \right) (Y_{c_d}R_{c_{d_1}} - Y_{c_{d_1}}R_{c_d})^2 < 0 \quad (vi).$$

In inequality (vi), the 1st, 2nd, and 4th terms are obviously positive. To satisfy the inequality, the 3rd term must thus be negative. However, because cluster d_1 and cluster # are subsets of cluster d , $Y_{c_d} > Y_{c_{d_1}}$ and $R_{c_d} > R_{c_{d_1}}$ are always true. Thus, the 3rd term is always positive, and inequality (vi) has no solution. It follows that the GLI monotonically decreases by the number of clusters.

Reviewer-only Figure D

[Comment]

6. In Figs 3 and 4, n=120 from 6 mice per group for the image analysis. Please defend this as an appropriate statistical approach. We don't use each slice in a microCT as a separate data point, so why is that done for each field here? The fields should be added or averaged for each mouse to get back to n=6. This may mean that many of the indices will no longer be statistically significant. Alternatively, you should at least visualize the datapoints with each mouse by color or symbol so the reader can see the within and between animal variability. In this vein, the word "indicated" on line 174 is too strong for the data shown. "Suggested" would be more appropriate.

[Response]

We thank the reviewer for this instructive comment. In response, we now show the data points for each mouse in color in the following reviewer-only Figure E, and we use the word "suggested" in the text:

Reviewer-only Figure E

“These results suggested that intermittent PTH treatment induced merged distributions of mOBs and mOCs with increased direct cell–cell contact between these two cell types.” (page 9, line 16-18).

[Comment]

7-1. It is hard to see how Fig 5h is n=6-8. Please explain.

[Response]

We have rewritten the legend of the revised Figure 5e as follows:

“(e) The duration of mOB–mOC contact. Data were collected in 7–8 independent experiments performed per group (control; $n = 322$, 1-w-PTH; $n = 375$, 3-w-PTH; $n = 1,252$, 6-w-PTH; $n = 1,126$).” (page 36, line 1-3).

[Comment]

7-2. Several statements in the discussion go beyond what has been demonstrated: Line 266 – there is no way to tell if the cells in contact area in the same or adjacent BMUs.

[Response]

We agree. We have removed the following text: “BMUs may influence adjacent

BMUs”.

[Comment]

7-3. Line 267 – causality has not been proven, only correlation.

[Response]

We totally agree and have rewritten the text as follows:

“Additionally, we found that the extent of direct contact of mOCs with mOBs was negatively correlated with the bone-resorbing activity of mOCs.” (page 13, line 18-19).

[Comment]

7-4. Line 286 – what if the osteoblast contact is with early, TRAP-negative osteoclast precursors? These would not be visualized here.

[Response]

We cannot completely exclude the possibility that mOBs in contact with early TRAP-negative osteoclast precursors promote osteoclastogenesis because we did not seek to visualize contact between such cells. In response to the reviewer’s concern, we have rewritten the text as follows:

“These results may suggest that direct cell–cell contact between mOBs and mOCs is not necessarily required for osteoclastogenesis and that soluble RANKL secreted from mOBs may also potently induce mOCs in vivo, although we cannot completely exclude the possibility that mOBs in contact with early TRAP-negative osteoclast precursors can also promote osteoclastogenesis.” (page 14, line 16-21).

[Comment]

7-5. Line 294 – the BRI was increased by a lot at 1 week, so how can you conclude that bone resorption was not increased.

It is possible that some effects of PTH are local and thus not reflected in the serum CTX. The microCT only shows net bone, so increased resorption early in some compartments such as calvaria (which was not assessed in the microCT) could be balanced by formation. Bone formation has not been analyzed in these mice in situ.

[Response]

We thank the reviewer for this instructive comment. As the reviewer pointed out, it was possible that bone resorption increased upon intermittent PTH treatment over 1 week, even though this was not reflected in any elevation in the serum CTX level. However, we did originally confirm that bone resorption was not increased by intermittent PTH

treatment for more than 3 weeks; we have rewritten the text to read as follows:

“We found that intermittent PTH treatment for more than 3 weeks significantly increased bone volume without enhancing bone resorption, despite a significant increase in the number of mOCs.” (page 15, line 3-5).

Response to Reviewer #3:

[Comment]

1. This manuscript describes studies carried out using a new technique to visualize living tissues. By using specific two-photon microscopy of transgenic mice the interactions of osteoclasts and osteoblasts could be followed in the mouse skull. Overall the manuscript is well written, the experiments appear well carried out and the data presented well. My only significant issue relates to the validation of the technique as it is important that the interpretation of the images is related to real cell interactions. Perhaps a simple *ex vivo* or *in vitro* experiment using the cells from the transgenic mice could easily demonstrate the accuracy of the imaging. Two-photon imaging could then be directly compared to normal imaging of the same cells at the same time. This would then validate the interpretation of the two-photon microscopy images. Overall the manuscript demonstrates an important step forward into our understanding of cell interactions.

[Response]

We thank the reviewer for this positive comment. As the reviewer pointed out, we performed *in vitro* experiments to investigate the interaction between mOBs and mOCs. To prepare osteoclasts, monocytes including osteoclast precursors were isolated from the bone marrow of Col2.3-ECFP/TRAP-tdTomato mice, cultured with M-CSF and RANKL for 5 days, and allowed to differentiate into mature osteoclasts (mOCs). To prepare osteoblasts, primary bone marrow cells were isolated from Col2.3-ECFP/TRAP-tdTomato mice and suspended in culture medium for 2 days to obtain bone marrow stromal cells (BMSCs). The BMSCs were cultured in osteogenic medium (with 50 μ M ascorbic acid and 10 mM β -glycerophosphate) for 14 days and thus differentiated into mature osteoblasts (mOBs). After mOBs and mOCs were thus prepared, the cells were co-cultured, and *in vitro* time-lapse imaging was performed (reviewer-only Figure F; Scale bar: 100 μ m). Thus, we

Reviewer-only Figure F

observed mOB-mOC contacts *in vitro*. However, the *in vitro* scenario differed from that evident *in vivo*, because of the many limitations of *in vitro* experiments. First, we could not directly isolate mOBs and mOCs from bones; this is very difficult. Second, the physiological microenvironment of the *in vivo* bone marrow (e.g., the cell density and cytokine levels) are not reproduced *in vitro*. Thus, we chose to perform *in vivo* imaging.

[Comment]

2. Line 47 abstract – the terminology “segregated fashion” might be better explained.

[Response]

We now use the word “discrete”:

“The mOBs and mOCs were distributed mainly in a discrete fashion, although some direct contact was detected in spatiotemporally limited areas.” (page 3, line 6-7).

[Comment]

3. Line 296 – The sentence beginning this line is difficult to understand.

[Response]

We agree. We have rewritten the text as follows:

“In clinical studies, weekly injections of teriparatide increased bone mass and the levels of bone formation markers without affecting bone resorption²⁴. However, the mechanism involved remains unclear. We found that intermittent PTH treatment for more than 3 weeks significantly increased bone volume without enhancing bone resorption, despite a significant increase in the number of mOCs. These phenomena may be attributable to inhibitory signaling from mOBs to mOCs mediated via direct cell–cell contact.” (page 15, line 1-6).

Reviewers' Comments:

Reviewer #1:

Remarks to the Author:

The authors have addressed most of the concerns raised in the previous review. In my opinion the manuscript has been improved significantly. The results are now more convincing with the help of the additional information on the image analysis. While there remain some concerns (see below), I believe this is an important paper and strongly recommend its publication. The ability to visualize mOB-mOC interaction and bone resorbing activity simultaneously is a technical tour de force. The novel findings that mOB and mOC in the steady state have a predominantly unmixed pattern of spatial distribution, whereas PTH treatment increases the mixing (and hence the cell-cell interaction), and that contact with mOB suppress mOC bone resorbing activity, are fine examples of the power of intravital microscopy to provide new biological insights.

Remaining concerns:

1. In response to the previous comment about "repulsive property", the authors have replaced with "a constitutive discrete property within a certain distance" and "discrete colony-forming exclusion with certain marginal zones". In my opinion, these statements are confusing and fail to describe the observation succinctly. I suggest something like "mOB and mOC tend to occupy distinct territories in the bone marrow in the steady state".
2. I appreciate that the time scale is different for the rapid R/N conversion of mOCs and the slow interaction between mOCs and mOBs. However, I believe it is important to resolve the conflicting logic that a) contact with mOB suppresses mOC resorption activity while b) mOC with low resorption activity (N type) are motile. How can motile mOC maintain contact with mOC? Shouldn't it be possible to perform time lapse imaging with short intervals and over long observation periods, thereby catching both fast and slow dynamics?

Reviewer #2:

Remarks to the Author:

Thank you for your very thorough response to all reviewers. All concerns have been addressed. I appreciate the analysis of the clustering methods and the decision to use a different clustering method as a result of the analysis. I would like the authors to add the comparison figure (Reviewer-only Figure B) in the supplementary materials. If the entire figure is too much, I would minimally like to see the results of Wards' algorithm under different perturbation of the images (i.e., first column of (c)) to validate its robustness. Similarly, I would like to see the proof of monotonicity (with Reviewer-only Figure D) appear in the supp. materials.

Reviewer #3:

Remarks to the Author:

The authors have largely addressed my concerns. However, I still find the first sentence of the abstract a little confusing with the use of semicolons and commas. In addition, I thank the authors for their explanation of why the in vitro methods were used. I suggest that this brief and simple explanation might be placed in the discussion so the reader can understand why these methods were chosen.

Print Email

Response to the Reviewers

We thank the reviewers for their positive and constructive comments on our manuscript (no. NCOMMS-17-03885-A), “**Dynamic modes of communication between mature osteoblasts and osteoclasts in vivo**” by Furuya *et al.* In response to the comments, we performed additional experiments/analyses and revised the manuscript. We hope that we have responded appropriately and that the revised manuscript is now suitable for publication.

Response to Reviewer #1:

[Comment]

1. In response to the previous comment about “repulsive property”, the authors have replaced with “a constitutive discrete property within a certain distance” and “discrete colony-forming exclusion with certain marginal zones”. In my opinion, these statements are confusing and fail to describe the observation succinctly. I suggest something like “mOBs and mOCs tend to occupy distinct territories in the bone marrow in the steady state”.

[Response]

We agree this point, and in response to the reviewer’s comment, we rewrote the text accordingly.

“This simultaneous visualization revealed that mOBs and mOCs mainly occupied discrete territories in the bone marrow in the steady state, although direct cell-to-cell contact existed in a spatiotemporally limited fashion.” (page 5, line 1-3).

“These imaging results revealed that mOBs and mOCs mainly occupied discrete territories and some direct cell-to-cell contact was detected in spatiotemporally limited areas.” (page 7, line 23-24).

[Comment]

2. I appreciate that the time scale is different for the rapid R/N conversion of mOCs and the slow interaction between mOCs and mOBs. However, I believe it is important to resolve the conflicting logic that a) contact with mOB suppresses mOC resorption activity while b) mOC with low resorption activity (N type) are motile. How can motile mOC maintain contact with mOB? Shouldn't it be possible to perform time lapse imaging with short intervals and over long observation periods, thereby catching both fast and slow dynamics?

[Response]

We greatly appreciate this constructive comment. We performed additional time-lapse imaging experiments with shorter time intervals to investigate the relationship between the motility of mOC and its contact with mOB (revised Supplementary Figs. 2h-n). After acquiring the images, we performed our 3D co-localization analysis to recognize contact of mOBs and mOCs at 10-min intervals (revised Supplementary Figs. 2h-k). In addition, we performed quantitative analysis of the motility of mOCs by using an image analysis software for tracking the time-dependent morphological changes as described previously (Kikuta et al., *JCI*, 2013). Briefly cell shapes were semi-automatically extracted by the software, and the cell deformation index (CDI) was calculated as the ratio of the cell areas changed within 10 min to the total cell area at $t = 0$. High or low CDI value correlates with the high or low motility of mOCs (revised Supplementary Fig. 2l).

As shown in revised Supplementary Figs. 2m and n, the CDI values of mOCs in contact with mOBs were significantly higher than those of mOCs lacking such contact, suggesting that mOBs increase the motility of mOCs (thus indicating inhibition of osteoclastic bone-resorptive activity) via direct cell–cell contact. We have added these new data to the revised Supplementary Figure 2, and we rewrote the text accordingly.

“In addition, we also analyzed the motility changes of mOCs in contact with mOBs. We have previously demonstrated that mOCs can be divided into two groups in terms of their motility and function, i.e., static resorbing osteoclasts (R state) and motile non-resorbing osteoclasts (N state)⁹. Here we performed 3D co-localization analysis and quantification of the motility of mOCs, and concordantly with the previous study, the motility of mOCs in contact with mOBs turned out to be significantly higher than those of mOCs lacking such contact (Supplementary Figs. 2h-n). These results also suggest that mOBs inhibit bone resorption of mOCs by direct cell–cell contact.” (page 8, line 16- page 9, line 1).

Response to Reviewer #2:

[Comment]

Thank you for your very thorough response to all reviewers. All concerns have been addressed. I appreciate the analysis of the clustering methods and the decision to use a different clustering method as a result of the analysis. I would like the authors to add the comparison figure (Reviewer-only Figure B) in the supplementary materials. If the entire figure is too much, I would minimally like to

see the results of Wards' algorithm under different perturbation of the images (i.e., first column of (c)) to validate its robustness. Similarly, I would like to see the proof of monotonicity (with Reviewer-only Figure D) appear in the supp. materials.

[Response]

We are very glad to receive this instructive response. In response to the reviewer's comment, we add these analyses data and the corresponding explanation in this article as 'Supplementary discussion'.

Response to Reviewer #3:

[Comment]

1. The authors have largely addressed my concerns. However, I still find the first sentence of the abstract a little confusing with the use of semicolons and commas. [Response]

In response to the comment, we have rewritten the text as follows:

“Bone homeostasis is strictly regulated by communication between bone-forming mature osteoblasts (mOBs) and bone-resorptive mature osteoclasts (mOCs). However the spatial-temporal relationship and mode of interaction in vivo have remained elusive.” (page 3, line 2-3).

[Comment]

2. In addition, I thank the authors for their explanation of why the in vitro methods were used. I suggest that this brief and simple explanation might be placed in the discussion so the reader can understand why these methods were chosen.

[Response]

We appreciate the reviewer's instructive comment. In response, we have added the following text in the discussion:

“Although we are also able to perform in vitro experiments and observe the interaction between mOBs and mOCs, the in vitro scenario differed from that evident in vivo, because of the many limitations of in vitro experiments. First, we could not directly isolate mOBs and mOCs capable of performing time-lapse imaging from bones. Second, the physiological microenvironment of the in vivo bone marrow (e.g., the cell density and cytokine levels) are not reproduced in vitro. Thus, we chose to perform in

vivo imaging.” (page 13, line 14-20).

Reviewers' Comments:

Reviewer #1:

Remarks to the Author:

The authors have done a thorough job addressing previous concerns, including conducting additional experiments showing a positive correlation between OB-OC contact and increased OC motility. It will be helpful if the authors could clarify, in the discussion section, how it is possible for OCs to maintain contact with OBs while increasing cell motility. No additional experiments are needed.

Response to the Reviewers

We thank Reviewer #1 for his/her positive and constructive comments on our manuscript (no. NCOMMS-17-03885-B), “**Dynamic modes of communication between mature osteoblasts and osteoclasts in vivo**” by Furuya *et al.* In response to the comment, we revised the manuscript. We hope that the revised manuscript is now suitable for publication.

Response to Reviewer #1:

[Comment]

The authors have done a thorough job addressing previous concerns, including conducting additional experiments showing a positive correlation between OB-OC contact and increased OC motility. It will be helpful if the authors could clarify, in the discussion section, how it is possible for OCs to maintain contact with OBs while increasing cell motility. No additional experiments are needed.

[Response]

We appreciate this constructive comment. In response to the reviewer’s comment, we rewrote the text accordingly.

“Most mOCs in contact with mOBs displayed dendritic shapes with synapse-like projections toward mOBs, and these projections moved actively while keeping contact with mOBs, leading to higher motility of mOCs.” (page 14, line 1-4).